

# An evaluation of liquid cloud droplet effective radius derived from MODIS, airborne remote sensing and in situ measurements from CAMP²Ex

Dongwei Fu[1], Larry Di Girolamo[1], Robert M. Rauber[1], Greg M. McFarquhar[2,3], Stephen W. Nesbitt[1], Jesse Loveridge[1], Yulan Hong[1], Bastiaan van Diedenhoven[4], Brian Cairns[5], Mikhail D. Alexandrov[5], Paul Lawson[6], Sarah Woods[6], Simone Tanelli[7], Ousmane O. Sy[7], Sebastian Schmidt[8,9], Chris Hostetler[10], Amy Jo Scarino[11]

[1]Department of Atmospheric Sciences, University of Illinois Urbana-Champaign, Urbana, Illinois, USA.
[2]Cooperative Institute for Severe and High Impact Weather Research and Operations, The University of Oklahoma, Norman, Oklahoma, USA.
[3]School of Meteorology, The University of Oklahoma, Norman, Oklahoma, USA.
[4]SRON Netherlands Institute for Space Research, Leiden, The Netherlands.
[5]NASA Goddard Institute for Space Studies, New York City, New York, USA.
[6]Stratton Park Engineering Company, Inc., Boulder, Colorado, USA.
[7]Jet Propulsion Laboratory, Pasadena, California, USA.
[8]Department of Atmospheric and Oceanic Sciences, University of Colorado Boulder, Boulder, Colorado, USA.
[9]Laboratory for Atmospheric and Space Physics, University of Colorado Boulder, Boulder, Colorado, USA.
[10]NASA Langley Research Center, Hampton, Virginia, USA.
[11]Science Systems and Applications, Inc, Hampton, Virginia. USA.

*Correspondence to*: Dongwei Fu (dfu3@illinois.edu)

**Abstract.** The cloud drop effective radius, *Re*, of the drop size distribution derived from passive satellite sensors is a key variable used in climate research. Validation of these satellite products often took place in stratiform cloud conditions that favored the assumption of cloud horizontal homogeneity used by the retrieval techniques. However, many studies point to concerns of significant biases in retrieved *Re* arising from cloud heterogeneity, for example, in cumulus cloud fields. Here, we examine data collected during the 2019 Cloud, Aerosol and Monsoon Processes Philippines Experiment (CAMP²Ex), which, in part, targeted the objective of providing the first detailed evaluation of *Re* retrieved across multiple platforms and techniques in a cumulus and congestus cloud region. Our evaluation consists of cross comparisons of *Re* between the MODerate resolution Imaging Spectroradiometer (MODIS) onboard the Terra satellite, the Research Scanning Polarimeter (RSP) onboard the NASA P-3 aircraft, and in situ measurements from both the P-3 and Learjet aircrafts that are all taken in close space-time proximity of the same cloud fields. A particular advantage of our approach lies in RSP's capability to retrieve *Re* using a bi-spectral MODIS approach and a polarimetric approach, which allows for evaluating bi-spectral and polarimetric *Re* retrievals from an airborne perspective using the same samples.

Averaged over all P-3 flight segments examined here for warm clouds, the RSP-polarimetric, in situ, and the bias-adjusted MODIS method of Fu et al. (2019) show comparable median (mean and standard deviations) of *Re* samples of 9.6 (10.2 ± 4.0) μm, 11.0 (13.6 ± 11.3) μm, and 10.4 (10.8 ± 3.8) μm, respectively. These values are far lower than 15.1 (16.2 ± 5.5) μm and 17.2 (17.7 ± 5.7) μm from the bi-spectral retrievals of RSP and MODIS, respectively. Similar results are observed when *Re* is segregated by cloud top height and in detailed case studies. The clouds sampled during CAMP²Ex consist of mostly small (mean transect length ~1.4 km) and low clouds (mean cloud top height ~ 1 km),





which are much smaller than the trade wind cumuli sampled in past field campaigns such as Rain in Shallow Cumulus over the Ocean (RICO) and the Indian Ocean Experiment (INDOEX). RSP bi-spectral *Re* shows larger relative values compared to RSP polarimetric *Re* for smaller and optically thinner clouds. Drizzle, cloud top bumpiness and solar-zenith angle, however, are not closely correlated with the overestimate of bi-spectral *Re*. We show that for shallow, non-drizzling clouds that dominate the liquid cloud cover for the CAMP²Ex region and period, 3D radiative pathways

appear to be the leading cause for the large positive biases in bi-spectral retrievals. Because this bias varies with the underlying structure of the cloud field, caution continues to be warranted in studies that use bi-spectral *Re* retrievals in cumulus cloud fields.

## 1 Introduction

Satellite retrieved cloud properties have been critical in advancing the understanding of the role of clouds in the

Earth's climate system. Still, the role of clouds in a changing climate remains a dominant source of uncertainty in climate change predictions (IPCC, 2013). Efforts to improve the accuracy of our satellite record of cloud properties continue to be called for (Ohring et al. 2005; NASEM 2018). This includes the record of cloud droplet effective radius (*Re*) of the drop size distribution. Satellite retrieved *Re*, owing to its wide spatial coverage and continuous monitoring record, has been applied for a wide range of studies such as estimating aerosol-cloud interactions (e.g., Menon et al.

2008; Ross et al. 2018; IPCC 2013) and evaluating model parameterizations (e.g., Ban-Weiss et al. 2014, Suzuki et al. 2013). By far the dominant approach for retrieving *Re* from space has been based on the bi-spectral technique of Nakajima and King (1990), which simultaneously retrieves cloud optical thickness (COT) and *Re* from visible/near infrared (VNIR) and shortwave infrared (SWIR) radiances. It has been applied to sensors such as the Advanced Very High-Resolution Radiometer (AVHRR, Rossow and Schiffer 1991), the Moderate Resolution Imaging

Spectroradiometer (MODIS, Platnick et al. 2003), and newer sensors such as the Visible Infrared Imaging Radiometer Suite (VIIRS, Cao et al. 2014) and the Advanced Himawari Imager (AHI, Bessho et al. 2016). Therefore, the longest records (spanning nearly four decades) of observations for cloud optical and microphysical properties are derived from the bi-spectral technique. Given its legacy and likely continued use in the future, it is essential to assess the error characteristics of the bi-spectral approach to advance the understanding of climate science, particularly as it applies

to cloud feedbacks (e.g., Tan et al. 2019) and aerosol-cloud interactions (e.g., Menon et al. 2008; Gryspeerdt et al. 2019).

There have been numerous studies aimed at understanding the error characteristics of *Re* retrieved from the bi-spectral technique. The largest errors are expected to occur whenever nature substantially deviates from the assumptions used by the bi-spectral technique, such as horizontally homogeneous clouds (hence, 1-D radiative transfer

as the forward model used in this retrieval), vertically homogeneous clouds, and a single-mode drop size distribution. Evaluations of *Re* from past field campaigns (e.g., Nakajima et al. 1991; Platnick and Valero 1995; Painemal and Zuidema 2011; McBride et al. 2012, Witte et al. 2018) show a ~ - 0.2 – 3 μm (~ - 2% - 40%) bias for MODIS and MODIS-like instruments, mostly for marine stratiform clouds under high sun conditions – conditions that are most favorable for the 1-D assumption (e.g., Loeb et al. 1998; Di Girolamo et al. 2010). 3-D radiative transfer simulations

suggest larger biases in the cumulus cloud fields that can reach ~100% (e.g., Marshak et al. 2006), with the bias closely



related to cloud heterogeneity and solar zenith angles. Under low sun conditions, Ahn et al. (2018) recently compared MODIS *Re* with airborne in-situ measurements over the Southern Ocean and reported a bias of 8 to 13 μm for non-drizzling clouds. A global perspective of the bias in MODIS *Re* was provided by Liang et al. (2015), who estimated zonal mean biases ranging from 2 to 11 μm by fusing data from MODIS and the Multi-angle Imaging

SpectroRadiometer (MISR, Diner et al. 1998). Their approach was further extended to regional estimates of the bias across the globe by Fu et al. (2019), which showed dependence of the *Re* bias on the cloud regime (i.e., larger bias in more cumuliform regimes). Fu et al. (2019) showed that the largest *Re* biases (up to +10 μm) occur over the tropical western pacific, which curiously is also the region where MODIS pixels detected as cloudy have the largest failures rates (up to 40%) in retrieving cloud optical and microphysical properties (Cho et al. 2015). Since liquid water clouds

in this region are dominated by cumulus and cumulus congestus clouds, a field campaign that in part targets the evaluation of *Re* retrievals for these clouds was warranted.

The Cloud, Aerosol and Monsoon Processes Philippines Experiment (CAMP²Ex; Di Girolamo et al. 2015), which took place in the Philippines and its surrounding waters from August to October of 2019, offers an opportunity for evaluating and understanding satellite derived cloud optical and microphysical properties in a heterogeneous

environment. Remote sensing and in-situ measurements of the clouds and aerosol fields were retrieved by the NASA P-3 and Learjet aircraft platforms. In this study, we focus on evaluating remotely sensed *Re* retrievals for warm cumulus and congestus clouds sampled during CAMP²Ex. Over the past several decades, satellite retrievals have not been evaluated in cumulus cloud fields, largely because of the difficulties in doing so. The fast-changing nature and complex cloud top structures of these clouds posed challenges for good cloud-top coordination between satellite

observations and airborne/in-situ measurements. CAMP²Ex provided tight coordination between Terra overpasses and the P-3 aircraft that carried the Research Scanning Polarimeter (RSP, Cairns et al. 1999). RSP provides bi-spectral and polarimetric retrievals of *Re*. The polarimetric *Re* is retrieved from multi-angle polarized radiances that are sensitive to single scattering. Past studies have indicated that the accuracy of polarimetric retrievals is less affected by the assumptions of plane parallel and homogeneous clouds than the bi-spectral technique (Bréon and Doutriaux-

Boucher, 2005; Alexandrov et al. 2012, Alexandrov et al. 2015). In this study, we rely on the RSP polarimetric *Re* to assess the RSP bi-spectral *Re* and MODIS *Re*. In addition, in situ derived *Re* from the P-3 and the Learjet platforms can also help to assess the performance of both the RSP retrieved *Re* and MODIS retrieved *Re*. There are several merits in cross-evaluating remotely sensed *Re* through comparison of data from different techniques and platforms: 1) RSP alone allows us to assess the performance of the bi-spectral technique against the polarimetric technique

without concerns on spatial and temporal collocation mismatches; 2) Comparing the MODIS bi-spectral *Re* against RSP bi-spectral *Re* can further assess the impact of measurement resolution (i.e., satellite vs. airborne) on the retrievals; and 3) P-3 in situ derived *Re* can assess the performance of the RSP polarimetric *Re* from the same airborne platform, whereas the Learjet in situ derived *Re* can further supplement the in situ derived *Re* from a different airborne platform. Along with RSP, the P-3 carried the High Spectral Resolution Lidar 2 (HSRL-2, Hair et al. 2008, Burton et

al. 2018), which provided measurements of aerosol properties and cloud top heights, and the Airborne Third Generation Precipitation Radar (APR-3, Durden et al. 2020), which provided precipitation information. Together they help to further investigate underlying relationships between the *Re* differences (difference between RSP bi-spectral



and polarimetric *Re*) and potential impact factors such as 3-D effects and drizzle. Thus, the objective of this study is to better understand the error characteristics of satellite retrieved *Re* and provide insights on future satellite instrumental designs by comparing bi-spectrally retrieved satellite *Re* with that from aircraft remote sensing and in situ measurements. In doing so, this study addresses the following questions:

1) What are the microphysical and macrophyscial properties of warm cumulus and congestus clouds sampled from a variety of observing systems during CAMP²Ex?

2) What are the relative errors between *Re* values retrieved from the bi-spectral techniques of MODIS and RSP, the bias-corrected MODIS *Re* technique of Fu et al. (2019), the RSP polarimetric technique, and in situ cloud probes?

3) How do these relative errors depend on factors such as cloud horizontal and vertical heterogeneity and drizzle?

This paper is structured as followed: In Sect. 2, the dataset and the methodology used in this analysis is presented. In Sect. 3, we first provide an overview of the sampled cloud's characteristics, and then examine the detailed behaviors of individual cloud fields, while focusing on the differences of the retrieved *Re* from different techniques. In Sect. 4, we further examine the dependence of the observed *Re* differences between the RSP polarimetric *Re* and bi-spectral *Re* on various impact factors (e.g., 3-D effects, sub-pixel heterogeneity, drizzle…), and discuss the consistency of representativeness of the *Re* retrieved from different techniques during CAMP²Ex. Finally, conclusion is provided in Sect. 5.

## 2 Data and methodology

### 2.1 CAMP²Ex dataset

The CAMP²Ex region was focused on the Philippines and its nearby waters, from approximately 6° N to 23 ° N, and 116° E to 128.5° E. 19 research flights of the NASA P-3 and 13 flights of the SPEC Learjet were flown during CAMP²Ex, 12 of which were joint missions. Sampled cloud fields include tropical storm convective cores, cold pools, broken shallow cumulus and congestus clouds. Frequent cirrus and altostratus clouds were also present during the flights. The P-3 platform was equipped with an array of instruments that included remote sensing instruments such as the RSP, HSRL-2, APR-3, and the SPN-S spectral pyranometer (Badosa et al., 2014). In situ probes such as the Fast Cloud Droplet Probe (FCDP, O'Connor et al. 2008) and 2-D Stereo Probes (2D-S, Lawson et al. 2006) were also installed on the P-3. The SPEC Learjet carried similar cloud microphysical probes as the P-3. There were 14 research flights (RF; Fig. 1) for the P-3 that were coordinated with Terra-MODIS overpass. Terra MODIS was chosen for the analysis rather than the Aqua MODIS or VIIRS is because the overpass time of the latter two sensors occurs in the afternoon when cirrus is more frequent and when the aircraft was returning to base that did not have favorable samplings. In addition, we applied the bias-adjustment technique of Fu et al. (2019), which was specifically developed for the Terra MODIS *Re*.



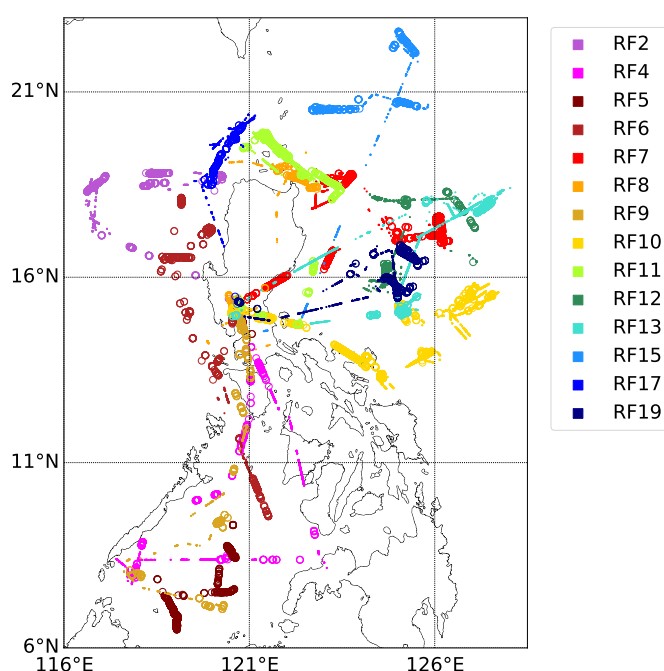

**Figure 1.** Flight tracks for 14 P-3 research flights with the Terra-MODIS overpass coordination over the CAMP$^2$Ex region. Dots indicate remote sensing legs with valid RSP retrievals; hollow circles indicate in situ legs with FCDP number concentration > 10 /cm$^3$.

**Table 1.** CAMP$^2$Ex P-3 research flights with successful coordination between P-3 and MODIS. RFs in bold indicates successful overlap between RSP sampling and MODIS.

| Flight Date (by UTC start) | No. | Geographic Regions |
|---|---|---|
| **2019-08-27** | **2** | NE South China Sea, W Luzon |
| **2019-08-30** | **4** | W, E Sulu Sea |
| 2019-09-04 | 5 | S Sulu Sea |
| 2019-09-06 | 6 | W Luzon, Mindoro Strait |
| **2019-09-08** | **7** | NE Luzon, far E of Luzon |
| **2019-09-13** | **8** | N,S Luzon, Lingayen Gulf |
| **2019-09-15** | **9** | W,S Sulu Sea |
| **2019-09-16** | **10** | Mt Mayon then NE into Philippine Sea |
| **2019-09-19** | **11** | N Luzon, transit along E Luzon |
| **2019-09-21** | **12** | Far E of Luzon |
| **2019-09-23** | **13** | Far E of Luzon, then S toward SE Luzon region |
| **2019-09-27** | **15** | Far NE of Luzon |
| **2019-10-01** | **17** | NW, N Luzon |
| **2019-10-05** | **19** | Far E of Luzon |



### 2.1.1 RSP cloud retrievals

RSP (Cairns et al. 1999) is a multi-angle multi-spectral polarimeter that provides along-track scans at up to 152 views between view zenith angles of about $\pm 60°$. It measures total and polarized reflectance at nine visible and shortwave infrared channels. RSP retrieves both polarimetric and bi-spectral $Re$. The RSP retrieves polarimetric $Re$ using polarized reflectance of the cloud bow with scattering angles ranging between 137 and 165 degrees. The shape of the cloud bow is dominated by the single scattering properties of cloud particles, which is less susceptible to uncertainties caused by 3-D radiative effects and aerosol loading (Alexandrov et al. 2012). The polarimetric technique uses a pre-calculated look-up table of single-scattering polarized phase functions with various $Re$, $Ve$ (effective variance) and scattering angles. Polarimetric $Re$ is retrieved by applying a parametric fitting to determine the relation between the phase function and the observed polarized reflectance. For the bi-spectral technique, like MODIS, the RSP uses the nadir reflectance at 865 nm (channel with negligible absorption by water) and at 1588 nm and 2260 nm (channels with strong absorption by water) to retrieve $Re$ and COT from a look-up table of pre-calculated reflectance of the two channels as a function of $Re$, COT and sun-view geometry. In this analysis we mostly focus on the bi-spectral $Re$ retrievals from the 2260 nm. Note the maximum $Re$ for both polarimetric and bi-spectral look-up tables is 30 μm. The RSP retrievals are reported at ~0.8 second intervals (~1.2 Hz), depending on aircraft platform altitude and air speed, resulting in spatial resolutions of ~120 meters during CAMP²Ex.

One of the merits of using RSP for this evaluation study is its capability to provide collocated polarimetric and bi-spectral $Re$ retrievals. Thus, the comparison between the RSP bi-spectral and polarimetric retrievals does not need to consider uncertainty resulting from sampling and collocation (a common issue with cross-platform comparisons). Using RSP retrievals alone provides a comparison between the bi-spectral and polarimetric retrieval techniques. The RSP polarimetric retrievals have been examined in other field campaigns, showing good agreement of better than 1 μm compared to in situ measurements in stratocumulus cloud fields (e.g., Alexandrov et al. 2018; Painemal et al. 2021). Here, we extend its evaluation in cumulus cloud fields sampled during CAMP²Ex.

RSP retrieves cloud top heights (CTH) using a multi-angle parallax approach (Sinclair et al. 2017). In addition, a simple cloud mask based on reflectance thresholds is reported, and RSP reports cloud top height retrievals whenever the cloud mask is valid. As we will show in Sect. 4.1, we also make use of valid (non-zero) RSP CTH retrievals to organize cloud properties in cloud elements, where a contiguous set of CTH retrievals is labeled as one cloud element. Mean and standard deviations of retrieved quantities belonging to a cloud element is computed. This further allows us to relate cloud properties to cloud macrophysics such as cloud length (characterized by RSP transect length), and cloud top bumpiness (characterized by standard deviation of CTH) at a cloud element level.

### 2.1.2 SPEC in situ measurements

The Stratton Park Engineering Company (SPEC) provided an array of in situ cloud probes for CAMP²Ex on the NASA P-3 and the SPEC Learjet. During CAMP²Ex, the NASA P-3 often targeted clouds using stacked tracks of in situ cloud legs below and cloud remote sensing legs above the cloud field, while the Learjet provided only in situ measurements. The Learjet was equipped with in situ instruments only and data from this platform are used to characterize the cloud microphysical properties. For this study, SPEC in situ instruments include the Fast Forward-



Scattering Spectrometer Probe (FFSSP; Brenguier et al. 1998), the Fast Cloud Droplet Probe (FCDP; O'Connor et al.
2008), and the two-Dimensional Stereo (2D-S) probe (Lawson et al. 2006). The FFSSP and FCDP are similar

scattering probes that retrieve droplet number concentrations from the forward scattering of a laser impinging on cloud
droplets and provide the droplet size distribution in 21 size bins ranging from 1.5 to 50 μm in diameter. The two probes
share the same electronics and differ slightly in the design of the probe tips to reduce shattering. The FFSSP was only
installed on the Learjet, whereas the FCDP was installed on both the Learjet and P-3. The 2D-S is an optical array
probe that uses two orthogonal laser-beams to record images of particles and nominally provides size distributions for

diameters ranging from 10 to 3000 μm. We combined the FCDP/FFSSP and 2D-S cloud droplet size distributions for
diameters from 1-1280 μm to cover cloud droplet and drizzle sizes. The "breakpoint" to combine the FCDP/FFSSP
and 2D-S particle distribution is fixed at 40 μm. Sensitivity tests were carried out using various breakpoints from 25
to 45 μm. We found that the choice of breakpoint does not introduce differences greater than 1 μm in the derived $Re$
for most 1 Hz sample used in this study. The FCDP/FFSSP and 2D-S number concentrations are combined at 1 Hz

temporal resolution. Only drop size distribution with total number concentrations greater than 10 cm$^{-3}$ and temperature
greater than 0 °C are included in this study following thresholds used to define warm cloud in previous studies (e.g.,
McFarquhar and Heymsfield 2001). The value of $Re$ from the combined size distributions is calculated as

$$R_e = \frac{\sum_{i=1}^{N} n_i\ r_i^3}{\sum_{i=1}^{N} n_i\ r_i^2}, \tag{1}$$

where $n_i$ is the number concentration (#/cm$^3$) for individual size bins, $N$ is the number of bins, and $r_i$ is the bin-center

radius.

The CAMP$^2$Ex data also archives an $Re$ product for full-length cloud passes computed from size distributions
summed from all samples belonging to the cloud pass. These size distributions use the FCDP/FFSSP, 2DS, and the
High-Volume Precipitation Spectrometer (HVPS, Lawson et al. 1993) to extend the size distribution out to 3-5 mm
(in diameter). The multiple probes' size distributions are stitched together using breakpoints that vary from different

cloud passes. When compared to our 1 Hz derived $Re$ using only FFSSP/FCDP and 2D-S, our cloud-averaged $Re$
compared favorably to the cloud pass $Re$ stored in the database: The median differences within 1 μm for both P-3 and
Learjet data across all flights, but with a smaller tail in the $Re$ distribution towards larger values – particularly for the
Learjet samples, which targeted deeper clouds compared to the P-3. While acknowledging this difference, we used
the 1 Hz derived $Re$ from the FFSSP/FCDP and 2D-S since it has a horizontal resolution similar to RSP retrievals at

1.2 Hz. The effects of precipitation on our understanding of RSP bi-spectral and polarimetric $Re$ retrievals are
examined here using coincident APR-3 airborne radar data discussed below.

### 2.1.3 Ancillary data

Apart from the RSP, other remote sensing instruments onboard the P-3 platform provided information about the
sampled cloud fields and the surrounding environment that may influence retrieval accuracy. For instance, cirrus

above the aircraft can lead to large biases in the bi-spectral retrieved cloud properties as their absorbing effect is not
modelled in the retrieval (e.g., Chang and Li 2005). To identify the presence of above-aircraft cirrus, we utilize the
measurements from SPN-S (airborne prototype spectral Sunshine Pyranometer, Norgren et al. 2021). SPN-S was
mounted on top of the P-3 for measuring downwelling spectral total and diffuse irradiances at wavelengths ranging

from 380 to 1000 nm. We derived direct beam transmittances at 860 nm with the assumption that the solar direct-beam is attenuated as prescribed by the Lambert-Beer law. Proper plane attitude adjustment has been applied to the SPN-S data (Bannerhr and Glover, 1991). By collocating the SPN-S transmittance with the cloud retrievals from the Advanced Himawari Imager (AHI) (temporal difference < 10 min and spatial difference < 5 km), we found that the collocated samples have a SPN-S transmittance of less than 0.95 when the AHI cloud phase flag indicates cirrus clouds. Thus, a direct beam transmittance of 0.95 is used to filter out possible above aircraft cirrus contamination.

The Airborne Third Generation Precipitation Radar (APR-3) is used to detect in-cloud drizzle in this study. The APR-3 is a Doppler, dual-polarization radar system operating at three frequencies (13, 35, and 94 GHz). It was mounted looking downward from the P-3 and performed cross-track scans, which covered a swath that is within the ±25° scan range. The 94 GHz channel's sensitivity to cloud liquid water has led to many studies using it to detect drizzle (e.g., Tanelli et al. 2008; Dzambo et al. 2019; Lebsock and L'Ecuyer 2011). In our analysis, we discovered that Version 2.3 of APR-3 contained numerous segments containing calibration errors that showed up as large along-track discontinuities in the background noise. This affected about 10% of the total APR-3 data and was therefore removed in our analysis.

The High Spectral Resolution Lidar (HSRL-2, Burton 2018) is a three wavelength lidar that makes measurements of the atmosphere at 355 nm, 532 nm and 1064 nm. It retrieves CTH, and aerosol properties such as extinction coefficient, backscatter and AOD. In our analysis, we take advantage of HSRL-2's capability of providing high resolution CTH at 2 Hz, to supplement RSP in providing cloud macrophysics characteristics of the CAMP²Ex sampled clouds. As we will show in Sect. 4.2.1, we also use HSRL-2 2 Hz CTH to investigate clear sky contamination for the RSP cloud element analysis.

All the instruments on the P-3 platform were temporally synchronized to the meteorological and navigation information provided by the National Suborbital Research Center (NSRC).

Compared to past field campaigns, one advantage of CAMP²Ex is the availability of the continuous monitoring from the Advanced Himawari Imager (AHI) on the Himawari-8 geostationary satellite. AHI provides moderate resolution (1 km) reflectances over the entire CAMP²Ex region at 10-minute intervals, this is important for post-campaign data processing since it provides a continuous view of a cloud field's evolution through each research flight.

## 2.2 MODIS cloud retrievals

The main goal of this study is to evaluate and understand the performance of bi-spectral $Re$ during CAMP²Ex, including those retrieved by satellites. The satellite $Re$ retrievals in this study come from MODIS onboard the Terra satellite. Terra is in a sun-synchronous orbit and has an equator crossing time at 10:30 AM. The $Re$ retrieved from the Terra MODIS represents the longest, single-platform, global record of $Re$. In our analysis, we used MODIS Collection 6.1 Level-2 Cloud Products at 1 km resolution (MOD06 V6.1; Platnick et al. 2018(a)). For $Re$ and COT, only the standard product from fully cloudy pixels were included, thus excluding partially cloudy pixel. Only liquid water clouds were considered based on the cloud phase flag provided in the MOD06 product. Only MODIS granules that overlapped with the CAMP²Ex sampling regions during individual P-3 research flights are included. In this analysis, we focus on the $Re$ and COT retrieved using the 0.86 μm and 2.1 μm channel since it is the most widely used and RSP



has a similar channel at 2.26 μm. Some recent studies have discussed the validity of comparing the MODIS 2.1 μm channel to the 2.26 μm channel from VIIRS, AHI and RSP (e.g., Platnick et al. 2018(b); Zhuge et al. 2021). It was pointed out that the inconsistency in the spectral response function of the two wavelengths can lead to differences of ~1-2 μm between the $Re$ derived from the two wavelengths, which is much smaller than the $Re$ bias estimates of up to 10 μm reported in Fu et al. (2019).

### 2.3 Bias-adjusted MODIS cloud retrievals

The MODIS $Re$ bias estimates presented in Fu et al. (2019) are also evaluated by comparing against the CAMP²Ex dataset. As a continuation of Liang et al. (2015), fused MISR L1B radiance data and MODIS L2 cloud $Re$ were used to retrieve COT at MISR 9 view angles. Liang et al. (2015) revealed that the COT retrievals show a local minimum around the cloud-bow scattering direction (~140°), and this feature was prominent throughout both MODIS cloud
COT values and COT retrieved from MISR. They showed that this minimum was attributed to an overestimate in the MODIS $Re$ product, and that the value of $Re$ bias could be estimated. Fu et al. (2019) further stratified 8 years of the fused MISR and MODIS data by MISR nadir $\tau$ and cloud heterogeneity, to produce regional estimates of MODIS $Re$ bias and bias-adjusted $Re$ at 2.5° resolution for the months of January and July. Here we apply the July regional correction factors from Fu et al. (2019) at 2.5° to the MODIS L2 granules over the CAMP²Ex domain to better compare
with $Re$ derived from other techniques under similar seasonal conditions. This allows one to test the robustness of the correction. The average of the July correction factors over the CAMP²Ex domain is ~ 0.6. The correction factors over this region range from 0.25 to 0.97 depending on latitude, $\tau$ and cloud heterogeneity. We are interested in evaluating the capability of regional bias corrections to capture the actual variability at its original resolution (i.e., MODIS 1 km retrieval) as we compare to field measurements from CAMP²Ex.

### 2.4 Matching technique

One major challenge for constructing the evaluation framework is the collocation between different platforms. In CAMP²Ex, the P-3 performed both remote sensing and in situ sampling during the same flight; simultaneous sampling from both methods is therefore not possible. Furthermore, CAMP²Ex targeted mostly cumulus and congestus clouds that have faster evolution and shorter lifetime when compared to stratocumulus clouds. A sawtooth flight pattern
commonly used in field campaigns targeting stratocumulus regions (e.g., Curry et al. 2000; Painemal and Zuidema 2011; Witte et al. 2018; McFarquhar et al. 2021; Redemann et al. 2021) was not employed during CAMP²Ex. However, while a strict point to point comparison is not achievable, we adopted the following approach to collocate MODIS, RSP and in situ measurements from a statistical standpoint.

A valid collocation between MODIS and the P-3 occurs based on a spatial and temporal matching criterion. For
the case-by-case comparisons presented in Sect. 3.3, all samples within the tightest rectangular box circumscribing the P-3 flight path that fell within a ±1.5-hour of the MODIS overpass time are included in the comparison. This time window was chosen based on the examination of all the 10-min AHI imagery and forward/nadir videos from the P-3 to maintain a balance between ensuring a significant number of samples and ensuring that the airborne remote sensing and MODIS observe the same cloud features. The sensitivity of our results to tighter temporal windows (e.g., 30-min





and 1-hour) was tested and did not alter the patterns observed in our results. Of the 19 P-3 research flights, there are
        14 research flights that had successful overlap with Terra-MODIS overpasses (Fig. 1 and Table 1).

        When comparing remotely sensed *Re* with in situ derived *Re*, one limitation lies in the simplified representation
        of clouds in the algorithms. Current passive remote sensing assumes clouds to be homogeneous in both the horizontal
        and vertical direction, but this representation of clouds is different from reality. In nature, clouds tend to have *Re*
profile that increase with height (e.g., McFarquhar et al. 2007; Arabas et al. 2009), although relatively constant in the
        horizontal direction at a given height level (e.g., Pinsky and Khain 2020; Zhang et al. 2011). The vertical variability
        of *Re* is often observed from in situ derived *Re* at various levels throughout a cloud. For remotely sensed *Re*, however,
        satellite retrieved bi-spectral *Re* is viewed as a vertically weighted *Re* with peak weighting near cloud top (e.g.,
        McFarquhar and Heymsfield 1998; Platnick 2000). For the polarimetric *Re* retrievals, the vertical weighting is more
strongly peaked and closer to the cloud top compared to the bi-spectral technique. This is because the polarimetric
        signature is dominated by single-scattering contributions, with a mean penetration optical depth of ~0.5 and negligible
        contributions from levels below optical depth ~3 from cloud top (Miller et al. 2018). Thus, to directly compare in situ
        retrieved *Re* with satellite or airborne remotely sensed *Re*, many studies have used in situ measurements at the cloud
        top to evaluate satellite *Re* (e.g., Painemal and Zuidema 2011; Witte et al. 2018; Gupta et al. 2021). This requires
determining the altitude of cloud tops during the in situ legs, which is simple for stratiform cloud with the aircraft
        performing sawtooth flight patterns at cloud top but not so for cumulus cloud. Here, we made use of all in situ
        measurements throughout various levels of the cloud fields. While we know the altitude in which the aircraft
        penetrated cloud, we do not have coincident measurements of collocated cloud top. We exclude in situ samples for
        which collocated AHI brightness temperature at 11 $\mu$m is below 273 K. This removes deeper convective clouds
sampled by the aircrafts that are not observed in the warm clouds sampled by passive remote sensing (i.e., RSP). Since
        the two airborne platforms are equipped with similar SPEC probes, despite the differences in the platform and
        sampling, the two in situ datasets serve to complement each other, providing additional information that is key to the
        evaluation of remotely sensed bi-spectral *Re*. We pay special attention to these sampling issues in our comparison of
        in situ measured *Re* with remotely sensed *Re*.

**3 Results**

        **3.1 General cloud characteristics of CAMP[2]Ex**

        We begin by providing an overview of some general cloud characteristics derived using the remote sensing data
        collected by the P-3 for all the research flights. Only oceanic liquid water clouds are included based on the RSP cloud
        top liquid index (van Diedenhoven et al. 2012). Cloud segments with cirrus overlying are removed based on SPN-S
transmittance < 0.95. Data here are organized into cloud elements Figures 2(a) and (b) show the probability distribution
        functions (PDF) and cumulative distribution functions (CDF) for mean *Re* and COT, respectively, retrieved from RSP
        polarimetric and bi-spectral (using the 2260 nm channel) techniques. While the COT distributions among the two
        techniques are in good agreement, the two *Re* distributions are quite different: the polarimetric *Re* distribution mode
        occurs at ~6 μm, whereas the bi-spectral *Re* mode occurs at ~12 μm. The median polarimetric *Re* is 7.0 μm, and the



median bi-spectral *Re* is 16.1 μm. The median COT is 3.5 for the cloud bow (COT retrieved using total reflectances and polarimetric *Re*) and 4.2 for the bi-spectral. Figure 2(c) provides the PDFs of RSP mean CTH and HSRL-2 mean CTH, with HSRL-2 CTH from 2 Hz samples. The RSP and HSRL-2 CTH distributions are in excellent agreement, both showing that ~60% of the cloud elements sampled have mean cloud tops < 1 km. Figures 2(d) and (e) show the PDFs of cloud element transect lengths and clear lengths (between cloud elements) derived from the RSP and HSRL-

2 CTH mask. We see that 50% of cloud elements sampled by RSP have transect lengths less than 0.6 km, with a mean length of 1.4 km, while 50% of HSRL-2 derived cloud elements have transect lengths less than 0.4 km, with a mean of 2.2 km. Both techniques show a mean clear length (i.e., spacing between cloud elements) of ~2.5 km. 50% of the clear lengths are less than 1 km and 0.4 km, respectively, derived from RSP and HSRL-2. The difference between the RSP and HSRL-2 cloud lengths distributions (particularly at small lengths) highlights HSRL-2's capability of

detecting small clouds because of its higher resolution (2 Hz vs 1.2 Hz for RSP) and detection sensitivity. We note that these clouds are much smaller than trade cumuli sampled during INDOEX (the Indian Ocean Experiment, Lelieveld et al. 2001) using a Multi-Channel Radiometer (MCR) and during RICO (Rain in Shallow Cumulus Over the Ocean, Rauber et al. 1997) using ASTER (Advanced Spaceborne Thermal Emission and Reflection Radiometer, Abrams et al. 2000), both of which had 50% of cloud area-equivalent diameters less than 2 km (McFarquhar et al.

2004; Zhao and Di Girolamo 2007). Given that the mean cloud area-equivalent diameter is approximately 1.1 times of a random linear transect (e.g., Barron et al. 2020), the clouds sampled by RSP and HSRL-2 during CAMP²Ex are much smaller than INDOEX's or RICO's trade cumuli. As such, the 1-km resolution MODIS pixels are expected to have a considerable amount of sub-pixel clouds during CAMP²Ex. We speculate that the reason for the maximum failure rate in MODIS cloud microphysical retrievals occurring over the western tropical Pacific, as reported by Cho

et al. (2015), may be the high frequency of small clouds here relative to anywhere else. Finally, Fig. 2(f) shows the PDF and CDF of the derived APR-3 W-band maximum reflectivity within individual RSP cloud elements. The APR-3 W-band maximum reflectivity median is at -9.24 dBZ. Past studies have shown a threshold of W-band column maximum reflectivity of ~0 dBZ is associated with high confidence of drizzle (e.g., Dzambo et al. 2019; Wang and Geerts 2003). From Fig. 2(f), 73% of the valid APR-3 W-band maximum reflectivity values are less than 0 dBZ,

indicating that most cloud elements sampled by RSP are not drizzling. Overall, Fig. 2 reveals that most clouds observed by the P-3 remote sensors are small, optically thin, non-drizzling. Most of the clouds were low clouds with tops under 2 km.  The samples exhibited a large difference (a factor of ~2) between RSP bi-spectral and polarimetric *Re* retrievals, which is investigated further in the sections below.



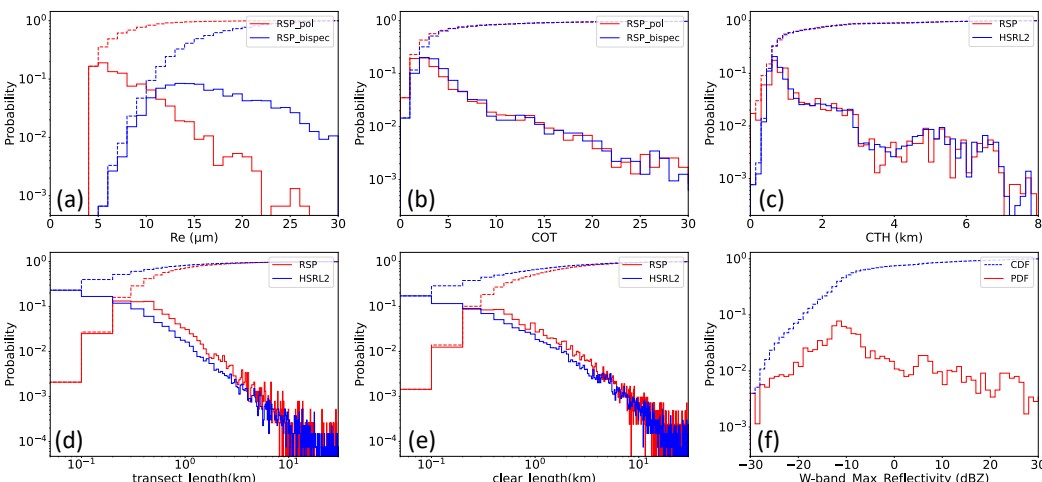


**Figure 2.** Probability distribution function (PDF, solid line) and cumulative distribution function (CDF, dash line) for cloud element mean values of (a) the RSP polarimetric and bi-spectral *Re,* (b) the RSP polarimetric and bi-spectral COT (c) CTH from RSP and HSRL-2, (d) cloud element transect length from RSP and HSRL-2, (e) clear segment length (between cloud elements) from RSP and HSRL-2, and (f) APR-3 W-band maximum reflectivity within a cloud

element.

## 3.2 RSP cloud microphysics statistics

The ability to retrieve both collocated polarimetric and bi-spectral *Re* from RSP allows us to compare the performance of the two techniques without further concerns on sampling differences. Figure 3 shows 2-d histograms of RSP polarimetric and bi-spectral *Re*, and *Re* differences (the difference between bi-spectral and polarimetric *Re*) as

a function of COT and CTH, using all 1.2 Hz samples passing the above P-3 cirrus filter for oceanic cloud samples during all flights. the differences between bi-spectral COT and cloud bow COT as a function of polarimetric *Re*, and cloud bow COT as a function of CTH are also shown. Several key features are displayed in Fig. 3. Figure 3(a) shows that most of the bi-spectral *Re* are larger than the polarimetric *Re*. A linear regression shows the correlation between the two *Re* is 0.38, with a bias (difference) of 6 μm, and RMSE of 8.2 μm. Figure 3(b) shows a rapid increase in *Re*

difference as retrieved optical depths decrease below 5. In other words, the largest *Re* differences are associated with optically thin clouds, which is consistent with the findings from the deployment of RSP during ORACLES (Miller et al. 2020). For CAMP[2]Ex, the differences between the two *Re* retrievals has a mean of 6.0 μm with a maximum of 26 μm, compared to mean difference of ~1 μm and maximum of 15 μm for ORACLES. The likely reason for the much larger *Re* differences in CAMP[2]Ex is the greater cloud heterogeneity in the oceanic regions around the Philippines

compared to stratocumulus cloud sampled in ORACLES. COT retrievals from the two techniques do not show large differences, as indicated in Fig. 3(c). Most of the COT differences are less than 2 (~20%), which is similar to the results in Miller et al. (2020). Finally, when the *Re* differences are binned by CTH (Fig. 3(d)), the *Re* differences decrease as CTH increases for low to mid-level clouds (CTH < 4km). As seen from Fig. 3(e), COT increases with CTH which would also result in liquid water path increasing with CTH. Beyond 4 km, no clear trend of *Re* difference

related to CTH is observed, perhaps because the population is largely alto-clouds as evident in Fig. 3(e).



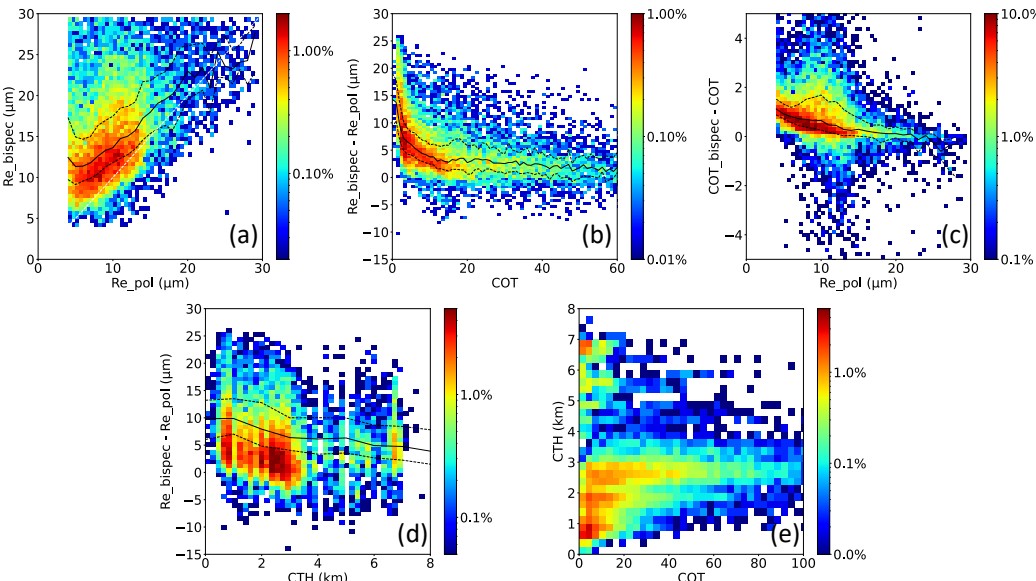

**Figure 3**. RSP 2-d density histogram of (a) polarimetric *Re* vs. bi-spectral *Re*; (b) RSP COT (cloud bow COT) vs. *Re* difference (bi-spectral *Re* – polarimetric *Re*); (c) polarimetric *Re* vs. COT difference (bi-spectral – cloud bow); (d) CTH vs. *Re* difference (bi-spectral *Re* – polarimetric *Re*); and (e) cloud bow COT vs. CTH. Black solid lines in (a) through (d) plot the median with respect to each horizontal bin, black dash lines indicate interquartile.

This comparison indicates that the bi-spectral *Re* are considerably larger than the polarimetric *Re*. However, without further examining the details of the macrophysics and the microphysics of the sampled cloud fields, it is difficult to comment on possible causes for the observed *Re* differences. Therefore, it is necessary to focus on individual cloud fields to study the characteristics of each cloud field (Sect. 3.3), and then relate the observed *Re* differences to other observed properties, such as cloud macrophysics and the presence of drizzle. Possible causes of the differences between the two *Re* retrieval techniques are further explored in Sect. 4.

### 3.3 Individual case studies

In our analysis above, we examined the general cloud characteristics and RSP bi-spectral and polarimetric retrieval differences over all 19 P-3 RFs from CAMP²Ex. Here we provide a few case studies to illustrate detailed inter-comparisons between remote sensing (satellite and aircraft) and in situ retrievals of *Re* during CAMP²Ex. Cases were selected when there was a good overlap between the sampling of MODIS, RSP and in situ over cirrus-free liquid-phase cloud fields over ocean. Table 2 provides the details of selected cases including the geolocation, MODIS overpass time, selected collocation time period of RSP, and in situ from P-3 and Learjet. For each case, we compare *Re* from RSP, in situ from the P-3 platform, in situ from SPEC Learjet, MODIS, and the bias-adjusted MODIS *Re* of Fu et al. (2019) to evaluate the performance of bi-spectral *Re* against polarimetric *Re* and in situ *Re* measurements. For a broken shallow cumulus case from RF17, the collocated hi-resolution ASTER (also onboard Terra) data allow us to highlight the representativeness of MODIS L2 cloud retrievals in sub-pixel cloud fields.



**Table 2**. Individual case details: Research Flight (RF) designation, approximate domain-center geolocation, MODIS overpass time (UTC), RSP, P-3 in situ and Learjet in situ time periods (UTC), and a brief description of sampled cloud fields.

| RF | Domain center lat/lon | MODIS overpass time | RSP time period | P-3 in situ time period | Learjet in situ time period | Features of interest |
|---|---|---|---|---|---|---|
| 2 | 18.5° N, 117° E | 8/27/19 3:05 | 8/27/19 3:36-4:30 | 8/27/19 3:00-3:30 | N/A | Field of shallow to moderate cumulus |
| 7 | 19° N, 123.5° E | 9/9/19 2:34 | 9/9/19 1:00-2:12 | 9/9/19 3:00-3:45 | 9/9/19 1:48-2:18 | Isolated cold pool/convective clouds |
| 12 | 18° N, 125° E | 9/22/19 2:03 | 9/22/19 2:12-3:18 | 9/22/19 1:50-2:05 | 9/22/19 1:00-1:42 | Moderate convection, cold pool and shallow cumulus |
| 17 | 20° N, 120° E | 10/2/19 2:40 | 10/2/19 1:55-4:10 | 10/2/19 1:10-1:50 | N/A | Field of small broken shallow cumulus |

### 3.3.1 27 Aug. 2019 research flight 2 – Terra clouds

During RF2, shallow convection was observed near 18.6° N, 116.9° E, as shown in the MODIS RGB image (Fig. 4(a)) during the Terra overpass at 03:05 UTC. The P-3 first entered the area depicted in Fig. 4(a) around 03:00 UTC on a low altitude leg (~500m) sampling below the shallow cumulus field. Between 03:00UTC and 03:30 UTC, the P-3 conducted several upward ascents into level legs to sample clouds in situ. Several high-altitude remote sensing legs were flown between 03:30 UTC and 04:30 UTC, sampling along a cumulus cloud line between 17° N to 19° N and 116° E to 117° E as indicated in Fig. 4(a). This cloud field occurred in the vicinity of a larger low-pressure system east of Luzon; some thin cirrus clouds are observed to the east of the sampled clouds. During the 1.5-hour time period, AHI imagery indicated that the shallow convective line retained its overall pattern and distributions, exhibiting consistent cloud top structures of typical broken shallow to moderate cumulus. Cirrus and ice clouds were filtered out from MODIS according to MODIS L2 phase flag. For the P-3 platform, the lower cumuli were mostly not affected by cirrus as seen from the AHI imagery and according to the SPN-s transmittance above the P-3. MODIS Level 2 retrievals show $Re$ ranging from 8 to 30 µm, associated with optically thin to moderately thick COT (1 - 50) and CTH of ~500m to 4000m. The RSP bi-spectral $Re$ also shows a range of 8 to 30 µm similar to MODIS. In great contrast, RSP polarimetric $Re$, bias-adjusted MODIS $Re$ and in situ derived $Re$ from P-3 all suggest a similar range of 5 to 15 µm (with only a few outliers ~20 µm), which is much lower than the bi-spectral $Re$ retrievals. The W-band maximum reflectivity from APR-3 indicates some precipitation in the deeper clouds (CTH > 2 km).

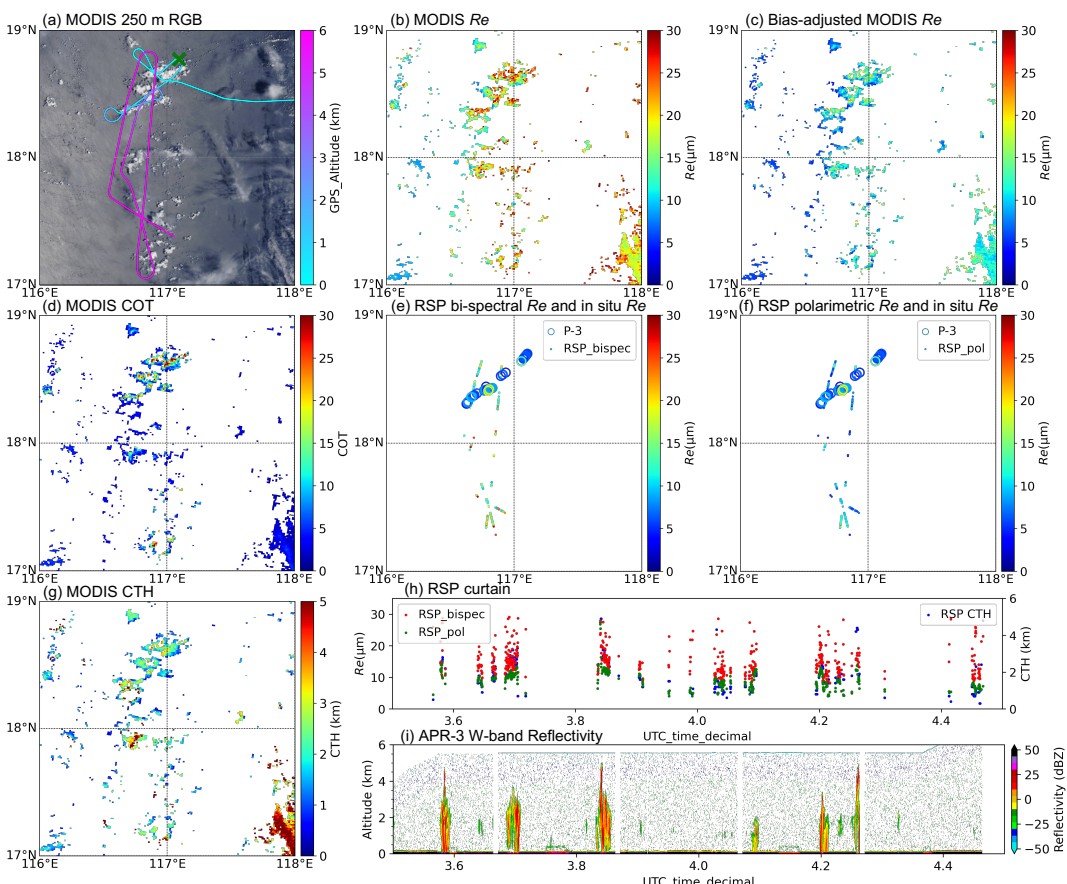

**Figure 4.** (a) MODIS RGB reflectance at 3:05UTC on 27 Aug. 2019. Color bar indicates P-3 altitude and flight track
within ±1.5 hours of MODIS overpass time. Green cross indicates P-3 location at MODIS overpass time. (b) MODIS
Level-2 1 km Re retrievals from 2.1 μm channel. (c) MODIS Level-2 1 km bias-adjusted *Re* retrievals from 2.1 μm
channel after applying Fu et al. (2019) correction factors. (d) MODIS Level-2 1 km COT from 2.1 μm channel. (e)
RSP bi-spectral *Re* retrievals from 2.26 μm channel. In situ *Re* from P-3 is displayed in circles. (f) RSP polarimetric
*Re* retrievals from 0.86 μm channel. In situ from P-3 is displayed in circles. (g) MODIS Level-2 1 km CTH retrievals.
(h) RSP *Re* and CTH curtain between 03:30 UTC and 04:30 UTC. (i) APR-3 W-band reflectivity between 03:30UTC
and 04:30UTC.

RSP bi-spectral and polarimetric *Re* have the same sampling, as does MODIS *Re* and the MODIS bias-adjusted *Re*.

So, they are directly comparable. But a direct pixel-to-pixel comparison between MODIS, RSP and in situ sampled

Re values is essentially impossible since they are not coincident in space and time. Still, the samples were collected

in a fairly small space and time window over which little overall change in the cloud field as indicated by AHI imagery.

Here, the *Re* retrievals were sorted into 250 m CTH bins. The *Re* mean and standard deviation are computed for each

height bin as a means of comparing remote sensing techniques' ability to capture the vertical variations of *Re*, which

is important when using the data for understanding cloud processes. Since the tops of cumulus clouds sampled in situ

are hard to determine, the platform altitude was used for in situ sampling, noting that these in situ derived *Re* are in-

cloud measurements rather than a vertically weighted *Re* as obtained from remote sensing. However, as noted by



Rosenfeld and Lensky (1998), *Re* is mostly conserved for a given temperature for non-drizzling clouds. Therefore, we binned *Re* retrievals from all 5 techniques (P-3 in situ, RSP polarimetric, RSP bi-spectral, MODIS bi-spectral, and bias-adjusted MODIS) separately as a function of binned CTH/altitudes. The results for the RF02 case are given in Fig. 5. All 5 techniques indicated an overall pattern of increasing *Re* with height. One prominent feature of Fig. 5 is

that for mid to low level cloud tops (below 3.5 km), the P-3 in situ (FCDP and 2D-S), RSP-polarimetric and bias-adjusted MODIS *Re* all indicate an increasing *Re* profile from ~7 μm to ~15 μm in the mean values. Thus, despite the differences in sampling and retrieval technique, the three are very consistent; the mean difference between the three *Re* profiles (Table 3) are all within 2 μm. The bi-spectral *Re* from RSP and MODIS, however, shows much larger values than the other three techniques, with increasing *Re* profiles from ~13 to ~22 μm. Thus, despite sampling and

resolution differences, these two bi-spectral products are consistent among themselves with the RSP bi-spectral *Re* ~3 μm smaller than that from MODIS. The bi-spectral *Re* from MODIS and RSP also show much greater *Re* variability at each height level (as seen from the horizontal whiskers), compared to RSP-polarimetric, in situ and bias-adjusted MODIS *Re*. For CTH below ~1.3 km, RSP bi-spectral *Re* suggests a decreasing *Re* profile, essentially opposite of other techniques. At higher altitudes, existence of drizzle tends to result in higher *Re* values with larger variability for

both RSP polarimetric and bi-spectral *Re* retrievals. The APR-3 and RSP curtains in Fig. 4 also confirms the correlation between drizzle and larger *Re* values for both techniques. At ~3.5 km, in situ derived *Re* indicated values of 13 - 20 μm, as it penetrated a convective cloud whose tops were higher than the P-3 by several hundred meters and visually appeared to be optically thick, as indicated by the P-3 forward video just before cloud penetration. Splashing of precipitation on the P-3 windshield was also evident from the forward video.

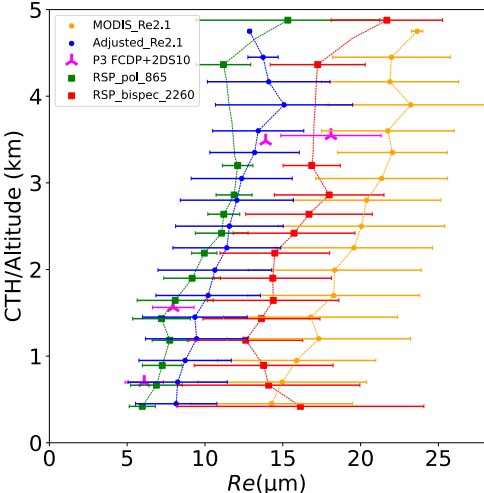

**Figure 5**. *Re* profile (mean *Re* vs. mean height) of vertically binned MODIS *Re*, bias-adjusted MODIS *Re* (after applying Fu et al. (2019) correction factors), RSP polarimetric *Re*, RSP bi-spectral *Re* and P-3 in-situ derived *Re* for the RF02 case. Horizontal whiskers indicate standard deviation of data within each 250m altitude bin.





### 3.3.2 09 Sep. 2019 research flight 7 – cold pool case 1

RF07 sampled the first cold pool targeted during CAMP²Ex. Around 01:00UTC Sep. 9[th], the NASA P-3 entered the cold pool region around 18.8° N, 122.8° E. The P-3 conducted multiple back and forth remote sensing legs at around 5.5 – 6.0 km altitude, targeting clouds along the cold pool front, followed by downward box spirals along and outside the front. The P-3 switched to in situ sampling near cloud base between 03:00 UTC to 04:00 UTC, finally exiting the region around 04:10 UTC. The SPEC Learjet provided additional in situ measurements as it conducted downward sampling of the cold pool and associated clouds from ~4.5 km to 0.4 km between 01:40 UTC and 02:20 UTC. A Terra MODIS overpass occurred at 02:47 UTC. At the MODIS overpass time, the P-3 was positioned to the west of the area shown in Fig. 6(a). As seen from the MODIS RGB reflectance image in Fig. 6(a), the cold pool at this stage became mature (as evident from AHI 10-minute imagery), with a clearly discernible convective line along the gust front, and a deeper (CTH ~12 km) convective cloud structure next to the cold pool. Here we again only focus on warm cumulus clouds, so only the cold pool clouds with liquid phase tops as indicated by the MODIS cloud flag are included. As indicated from MODIS L2 CTH, the congestus and deeper convective clouds have cloud tops around 3 to 6 km altitude, with shallower clouds along the gust front. The MODIS $Re$ shows a range of 8 to 20 μm for the shallower clouds (with optical depth 1-15), and 20 to 30 μm for the deeper clouds (with optical depth > 40). Again, this is closely in line with the RSP bi-spectral $Re$ in the range of 7 to 30 μm. Bias-adjusted MODIS $Re$ and polarimetric $Re$ agrees on the range of 4 to 25 μm. The P-3 in situ $Re$ values are in the ~5 to 6 μm range, whereas Learjet in situ $Re$ suggests a range from 10 μm to beyond 30 μm. This large contrast between Learjet and P-3 was primarily due to different sampling strategies: The Learjet entered the cold pool convective core at an earlier stage (~1:50 UTC), sampling through the top of the convective clouds ~4.5 km that was heavy precipitating, as large splashing on the Learjet windshield was observed according to the forward video. The P-3, however, sampled near cloud base (~0.5 km) at a much later stage (~3:30 UTC) as clouds start to dissipate (as observed from the AHI 10 min imagery). No clear drizzle was observed from the P-3 forward video. The APR-3 W-band maximum reflectivity also indicates considerable precipitation (maximum reflectivity ~25 dBZ in Fig. 6 (h)) during the time periods which was sampled by RSP and Learjet.

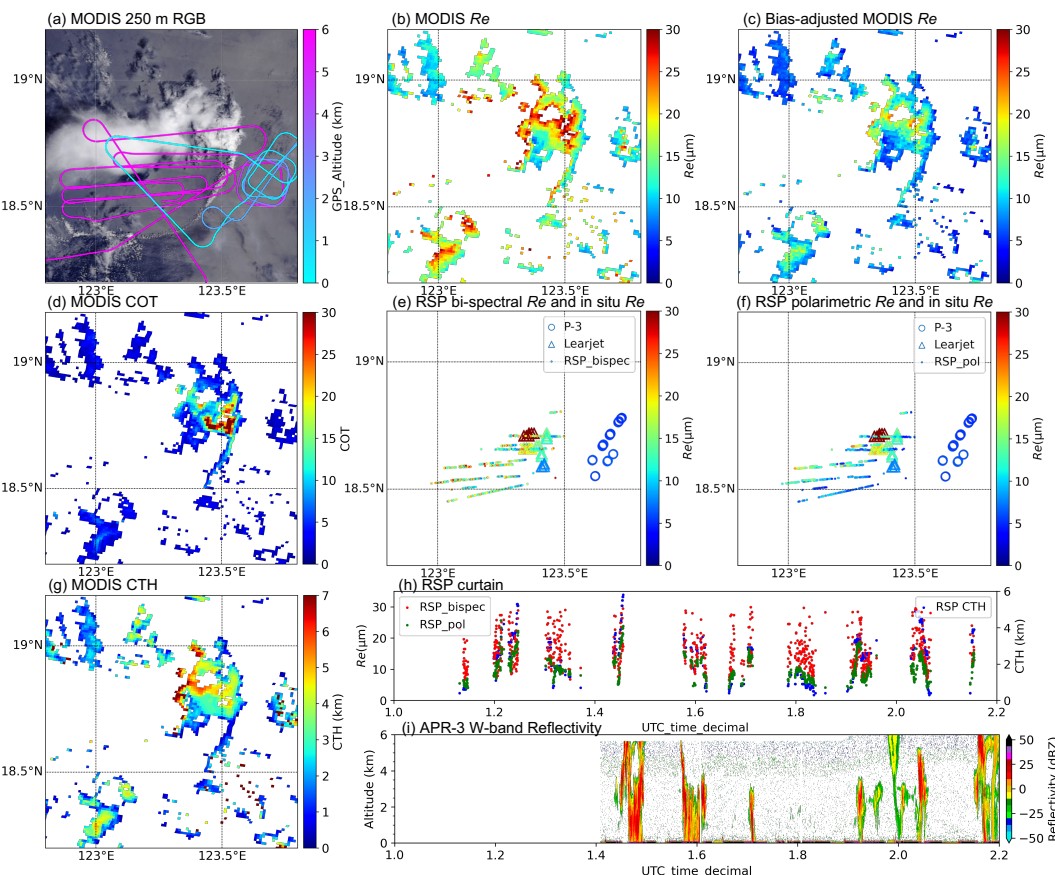


**Figure 6.** Same as Fig. 4, but for RF7 on 09 Sep. 2019.

Figure 7 is constructed using the same approach as Fig. 5 and shows an increasing *Re* profile with height for all six techniques, but with significant differences. While the RSP polarimetric *Re* shows a clear increasing trend up to ~3.5 km, the RSP bi-spectral *Re* shows much more variability in the *Re* mean values throughout various CTH bins.

Both MODIS profiles (original and bias-adjusted) exhibit a small increasing trend of *Re* profile below ~2 km, above which the trend becomes larger. The in situ profile from the Learjet also shows a clear increasing *Re* profile. When the Learjet in situ *Re* profile is compared to the remote sensing *Re* profiles, the difference between the two becomes more prominent at higher altitudes. In situ measurements may be penetrating through deeper convective clouds than those sampled by remote sensing with CTH of similar altitude. As indicated in Fig. 7, for the shallower clouds below

2 km the Learjet and P-3 in situ *Re* are in very good agreement with the RSP polarimetric *Re* (e.g., mean bias between RSP polarimetric and Learjet in situ *Re* is 0.8 μm), but as the height exceeds 2 km, Learjet-derived *Re* mean values exceed 20 μm and increase to 40 μm at ~ 4.5 km altitude. These very large *Re* values are associated with the heavy precipitation observed during the RF07. According to the Learjet forward video, at ~1:47 UTC the Learjet penetrated through the side of a raining congestus that is close to the convective core at approximately ~ 4.5 km in altitude. Heavy

splashing on the Learjet windshield was observed from the forward video. The Learjet then descended while





penetrating through raining clouds (indicated by apparent splashing on windshield through the forward video) until ~ 2:10 UTC at an altitude of ~1 km. RSP polarimetric *Re* appears to be in good agreement with bias-adjusted MODIS *Re* with a mean bias of 1.6 μm. Again, MODIS *Re* and RSP bi-spectral *Re* appear to be much larger than RSP polarimetric *Re* (mean bias between RSP polarimetric *Re* and MODIS *Re* / RSP bi-spectral *Re* is 6.1 μm / 5.0 μm).

The abundance of precipitation in this cloud scene for higher cloud-top clouds (Fig. 6(j)) leads to larger *Re* retrievals from both RSP polarimetric and bi-spectral techniques that are closer between each other at altitudes between 3 - 4.5 km.

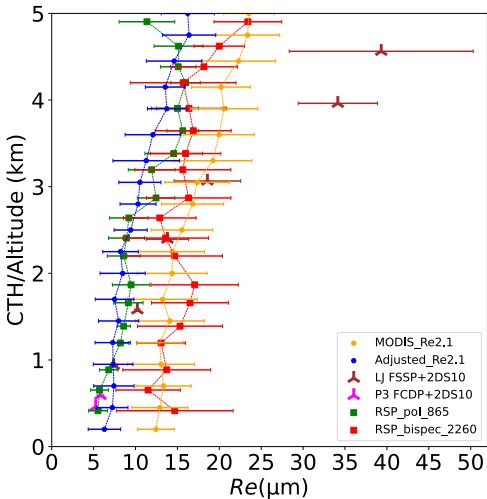

**Figure 7.** Same as Fig. 5, but for RF07 and the additional Learjet in-situ *Re* values.

**3.3.3 22 Sep. 2019 research flight 12 – cold pool case 2**

RF12 observed another cold pool system. At around 1:40UTC on Sep. 22nd, the P-3 entered the scene depicted in Fig. 8 on a ~0.7 km low altitude leg, where it then started to perform back and forth in situ measurements near the cloud base at the cold front region. Around 02:00 UTC, the P-3 platform started to ascend in an upward spiral and then switched into remote sensing legs at ~4.5 km altitude as it flew westward to sample cumulus turrets. After

repeated remote sensing legs were conducted, the P-3 left the scene at ~03:15 UTC, exiting the right of the domain in Fig. 8(a). The Learjet entered the scene at ~01:00 UTC at approximately 5 km altitude. As it flew westward, it gradually spiraled to cloud base to sample the lifecycle of the cumulus turrets until ~01:42 UTC, when it exited the region at the bottom left of the scene. The Learjet forward video indicates that the platform encountered precipitation as it penetrated throughout the cumulus cloud field. Terra MODIS overpass at 2:03 UTC observed the cold pool in its

mature stage (as evident from AHI). The MODIS RGB image (Fig. 8(a)) shows cirrus clouds to the south and east and north of the cold pool system. MODIS liquid cloud *Re* shows a range of ~8 to 25 μm, with COT ranging from ~1 to 40. RSP bi-spectral *Re* suggests a similar range of 7 to 30 μm. The RSP polarimetric and P-3 in situ *Re* shows *values of* ~4 to 10 μm, and the bias-adjusted and Learjet in situ *Re* shows slightly higher values of 5 to 15 μm. APR-3 curtain suggests precipitation for the deeper convective clouds (Fig. 8(j)).

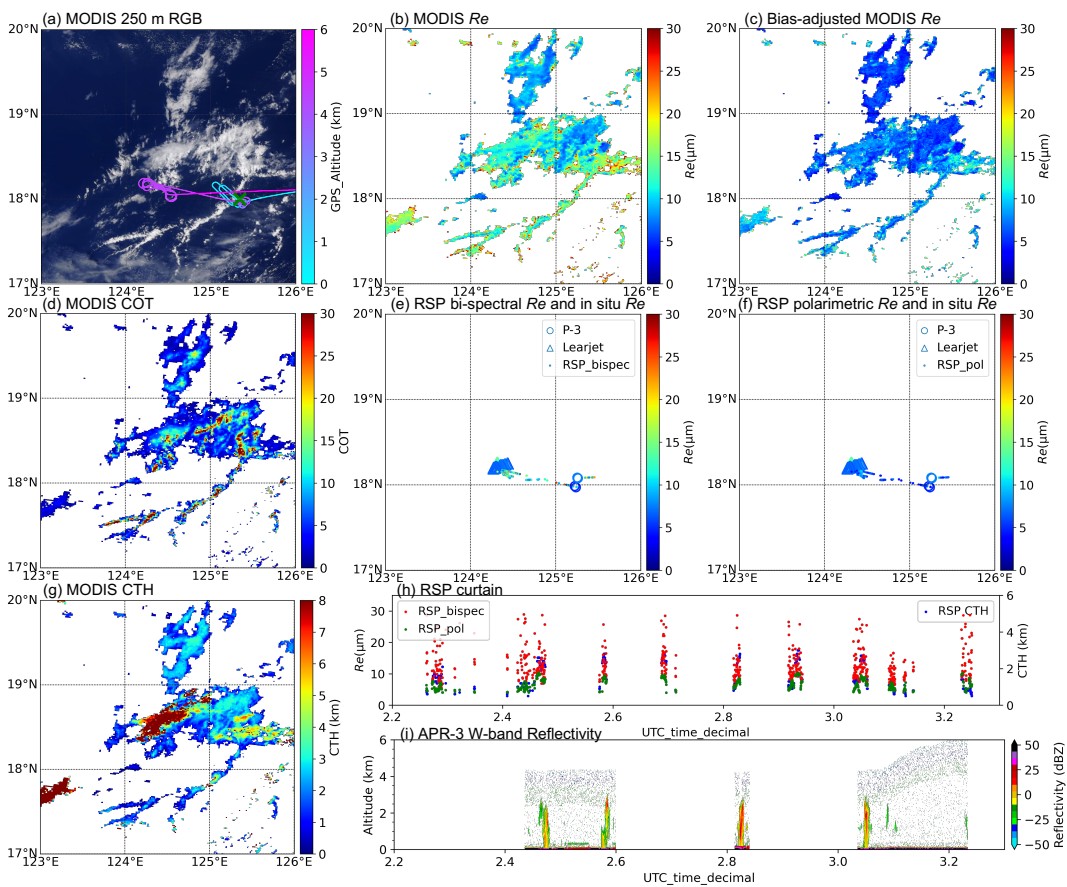

**Figure 8.** Similar with Fig. 4, but for cold pool case 2 on 22 Sep. 2019.

Figure 9 shows much smaller vertical variations of *Re* with height when compared to the previous two cases. The

in situ derived *Re* from the Learjet and P-3 line up with each other. They both agree with RSP polarimetric *Re* especially below 2 km (mean bias ~ 1 μm). The Learjet derived *Re* above 3 km shows slightly larger *Re* mean values of ~14 μm resulting from the precipitation within the cumulus cloud field. Throughout all CTH levels, bias-adjusted *Re* is in good agreement with RSP polarimetric *Re* (mean bias ~ 1.6 μm). The MODIS *Re* and RSP bi-spectral *Re* both suggest much larger mean *Re* values. Their mean biases with respect to RSP polarimetric *Re* are 6.0 μm for the RSP

bi-spectral and 7.2 μm for the MODIS, respectively.



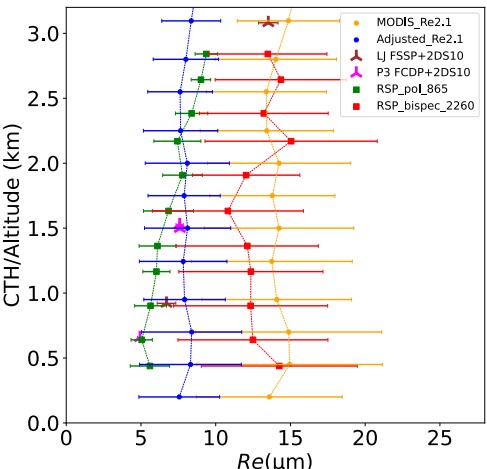

**Figure 9.** Same as Fig. 7, but for the RF12.

### 3.3.4 02 Oct 2019 research flight 17 – small broken shallow cumulus case

RF17 sampled a field of small, shallow cumuli that appear very different than the previous three cases. The most
prominent feature for this case is the abundance of small broken cumulus clouds in this domain (more discussion on
this later in this section). On October 2$^{nd}$ around 01:00UTC, the P-3 platform entered the region at around 1.5 km
altitude and then descended to ~100 m above sea level to begin in situ measurements below cloud base and at various
levels within clouds. Around 02:00 UTC, the P-3 started climbing from 1 km to 5 km altitude to perform remote
sensing sampling, with long stretches of straight legs as shown in Fig. 10(a). The aircraft exited the region around
04:30UTC. A MODIS overpass took place at 02:40UTC. The MODIS retrievals indicate that clouds in this case were
very shallow, broken (CTH below 1.5 km) and optically thin (COT below 10, mostly between 1 to 4), with $Re$ values
between 10 to 30 μm. Like MODIS, the RSP bi-spectral retrievals show a range of 6 to 30 μm. RSP polarimetric $Re$,
however, show much smaller values of 4 to 7 μm that also agrees with the P-3 in situ $Re$. Bias-adjusted MODIS $Re$
also shows a similar $Re$ range of ~5 to 10 μm. Only some slight drizzle was observed for the cloud ~1 km from the P-
3 forward video.



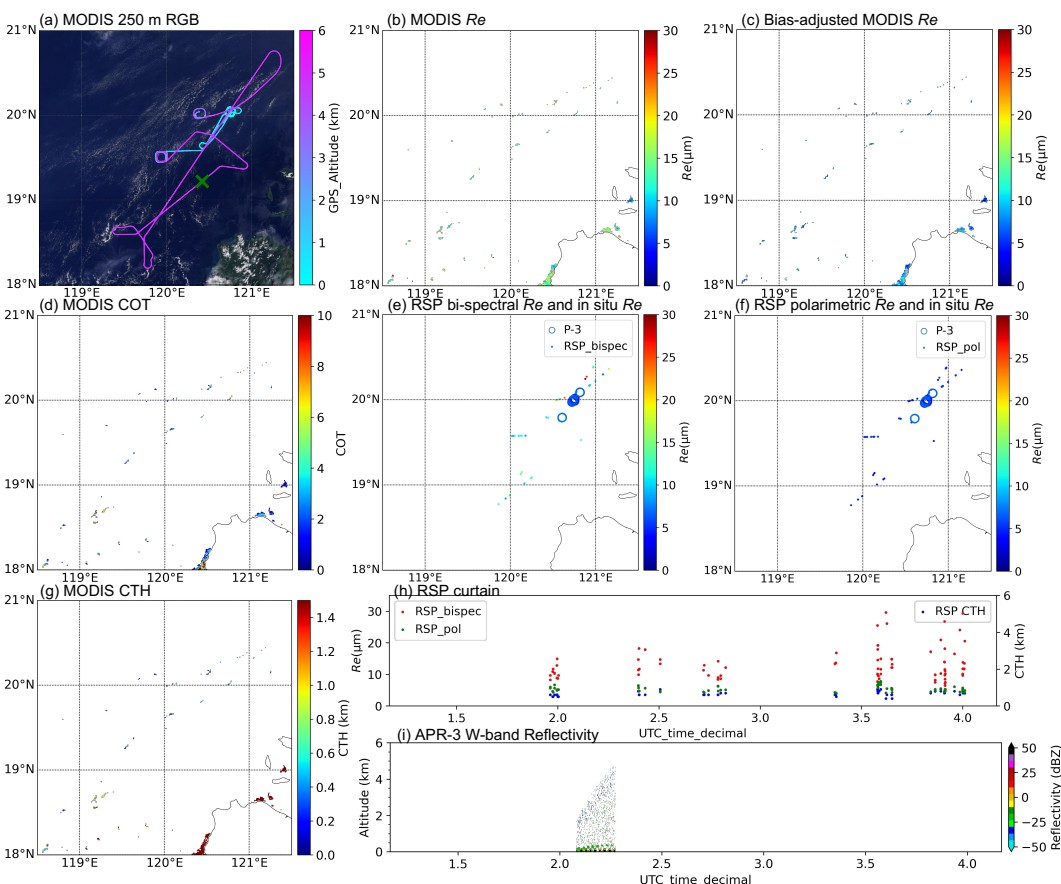

**Figure 10**. Similar with Fig. 8, but for broken shallow cumulus case on 02 Oct. 2019.

Figure 11 shows that the *Re* profiles of P-3 in situ and RSP polarimetric *Re* are very consistent, both suggesting an increasing *Re* profile with height (with a range of 5 - 7 μm). The mean bias between RSP polarimetric and P-3 in situ is 0.1 μm. The MODIS Re and RSP bi-spectral *Re* also share similar *Re* mean values (13 to 16 μm), with the RSP bi-spectral *Re* showing much more variability. The MODIS bias-adjusted *Re* are close in values with the in situ and RSP polarimetric *Re* values, but biased high by ~2 to 3 μm (mean bias between RSP polarimetric and bias-adjusted MODIS Re is 2.6 μm). Figure 11 shows the two bi-spectral *Re* profiles are much larger than the other three *Re* profiles, e.g., the mean biases with respect to RSP polarimetric for RSP bi-spectral *Re* and MODIS *Re* are 9.8 μm and 8.3 μm, respectively.





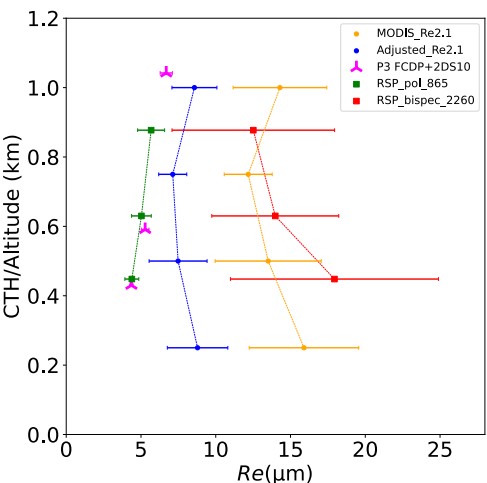

**Figure 11.** Same as Figure 5, but for the RF17 broken cumulus case.

To better demonstrate the representativeness problem with MODIS retrievals in this common type of cloud field, we overlayed the ASTER 15 m radiances over the selected RF17 cloud scene with the MODIS 1 km $Re$ retrievals (Fig. 12). Figure 12(a) shows that while there are numerous very small broken cumulus clouds in the scene, the MODIS Level 2 $Re$ only reported a handful of pixels having successful retrievals. Figure 12(b) is a zoomed in view of the ASTER 15 m granule overlayed with the MODIS $Re$. It clearly shows that these small cumuli are sub-pixel for MODIS Level 2 retrievals, and that the cloud variability cannot be resolved by MODIS. It is expected that this unresolved variability leads to biases in MODIS retrieved $Re$, which we further investigate in Sect. 4.2. While not shown, many clouds in the scene are correctly identified by MODIS as partly cloudy pixels (PCL) and excluded from the standard $Re$ product analyzed here. Apart from these clouds being sub-pixel to MODIS retrievals cannot resolve, some failed retrievals may be attributed to the finite range of the bi-spectral look-up table (LUT). For example, Cho et al. (2015) showed that failed retrievals in the MODIS product would occur whenever the retrievals fall outside the LUT range, and this failure rate can be as high as 40% in the Southeast Asia oceanic region. This questions the validity of the representativeness of long-term MODIS climatologies for regions dominated by small cumulus cloud fields.

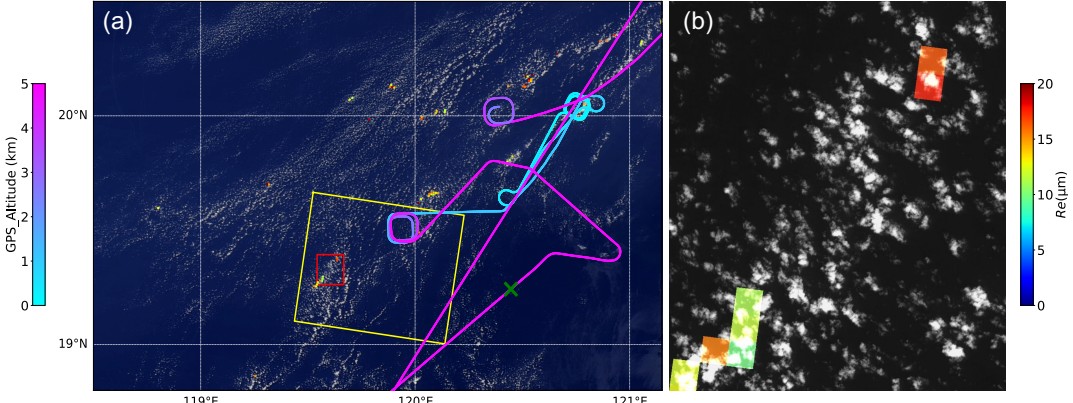

**Figure 12.** (a) MODIS 250 m RGB Reflectance at MODIS overpass (2:40UTC) overlayed by MODIS Level 2 liquid Re retrievals, P-3 flight path within 1.5 hours of MODIS overpass color-coded by altitude. The green cross indicates P-3 position at MODIS overpass time. Yellow box indicates outline of the collocated ASTER 15 m granule. Red box indicates outline of the zoom-in view. (b) zoom-in view of ASTER 15 m resolution 3N channel radiance, overlayed by MODIS Level 2 liquid Re retrievals.

### 3.3.5 Summary for four cases

Table 3 summarizes the mean difference between the RSP polarimetric Re with the other techniques for the four cases discussed above. At lower altitudes, the Learjet and P-3 in situ and RSP polarimetric Re are in good agreement. While acknowledging the definition of Re is different for remote sensing retrievals and in situ measurements, the overall good agreement between polarimetric and in situ Re should bring more confidence in the performance of polarimetric Re retrievals in cumulus cloud fields. In all comparison cases, RSP polarimetric Re also agrees well with the MODIS bias-adjusted Re, despite the resolution and sampling differences between the two. Finally, both the RSP bi-spectral and MODIS Re show good agreement with each other, but with the overall largest Re values among all techniques. The distinct separation between bi-spectral Re and Re from all other techniques implies an overestimate in bi-spectral Re for cumulus clouds regions; the mean Re differences of ~5-10 µm in the individual cases matching with the estimates of ~ 6 - 9 µm bias in 2.1 channel MODIS Re found by Fu et al. (2019).

The impact of drizzle on Re from these case studies were evident in remotely sensed and in situ observations. Direct comparison between in situ and remote sensing in deeper clouds containing drizzle was hampered by the fact that the in situ samples containing drizzle and large Re occurred at locations that were not close to cloud top according to the aircraft forward video. We therefore take a more extensive examination of the impact of drizzle on our comparison of bi-spectral and polarimetric retrievals in Sect. 4.2.2 using APR-3.

As mentioned in Sect. 2.2, we acknowledge the difference between the MODIS 2.1 µm channel to the RSP 2.26 µm channel, we acknowledge that the differences in the two wavelengths and we do not expect the Re retrieved from





MODIS and RSP bi-spectral to have the exact same bias. In our analysis, other contributing factors that may impact the differences between RSP 2.26 μm bi-spectral $Re$ and MODIS 2.1 μm $Re$ include (1) sampling differences (2) channel differences in the face of vertical and horizontal variations in cloud optical properties, and (3) pixel size difference in the face of 3-D variations. Despite these factors, the RSP 2.26 μm bi-spectral $Re$ and MODIS 2.1 μm $Re$

have very similar behavior, exhibiting a large positive bias and much greater variability in $Re$ relative to the other techniques.

**Table 3.** Mean bias (difference) between RSP polarimetric with the other techniques for the four cases discussed in Sect. 3.3.

| mean bias wrt. RSP Polarimetric $Re$ (μm) | RF02 | RF07 | RF12 | RF17 |
|---|---|---|---|---|
| RSP bi-spectral $Re$ | 6.0 | 5.0 | 6.0 | 9.8 |
| MODIS $Re$ | 9.2 | 6.1 | 7.2 | 8.3 |
| MODIS-Bias-adjusted $Re$ | 1.4 | 1.6 | 1.6 | 2.6 |
| P-3 in situ $Re$ | 0.5 | 0.2 | 0.8 | 0.1 |
| LJ in situ $Re$ | N/A | 9.4 (all) 0.8 (below 2 km) | 1.1 | N/A |

## 4 Relating Re bias to 3-D factors, sub-pixel heterogeneity and drizzle


Since much of the observed $Re$ bias between the bi-spectral and polarimetric technique occurs for small retrieved optical thickness (COT <5, Fig. 3(b)), it is especially important to consider the uncertainties in the retrieval process that can affect the bi-spectral retrieval, even when the core assumptions of 1-D radiative transfer are met. Sources of uncertainty include instrument calibration, atmospheric correction, surface Bi-directional Reflectance Distribution

Function (BRDF) and assumed size distribution shape as well as retrieval logic. These uncertainties are derived for MODIS in Platnick et al. (2018a), and globally validated for COT retrievals of oceanic liquid water clouds in the limit of homogeneous clouds in Di Girolamo et al. (2010). Still, even in the case of ideal simulated 1-D retrievals, retrieved $Re$ values can be biased high due to the presence of multiple $Re$ solutions and limitations of lookup table interpolation (Miller et al. 2018). However, these large $Re$ retrievals are much more frequent in our data (e.g., Fig. 3(b)) that can be

reasonably explained by these sources of uncertainty, as discussed below.

For MODIS, the uncertainty in retrieved $Re$ can be significant (16-30%) for (COT < 5) (Platnick et al. 2018(a)) Similar uncertainties can be anticipated for RSP, since RSP has a calibration uncertainty of 3% (Knobelspiesse et al. 2019) that is similar to MODIS. Even in the worst case that all uncertainties are systematic across the field campaign period, these uncertainties are much smaller than the factor of two differences observed between RSP polarimetric

and bi-spectral $Re$ with COT < 5. This indicates that other retrieval assumptions should be investigated to understand the cause of the observed differences between $Re$ retrieval techniques.

The literature contains extended discussions relating passive cloud retrieval bias in COT and $Re$ to the impact of sub-pixel heterogeneity and other 3-D effects (e.g., Marshak et al. 2006; Zhang and Platnick 2011; Zinner et al. 2010; Zhang et al. 2012), sun-view geometry (e.g., Loeb and Davies 1996; Várnai and Davies 1999; Liang and Di Girolamo

2013; Grosvenor and Wood, 2014), and the presence of drizzle (e.g., Zhang et al. 2012; Ahn et al. 2018). In this section, we test the hypothesis that retrieval errors from 3-D effects contribute to measurable bias in $Re$ retrievals from CAMP²Ex dataset. To facilitate our investigation, we used a cloud element labelling technique based on RSP L2 cloud



retrievals, where a cloud element is defined as a region with contiguous CTH retrievals. We then provide statistics of cloud microphysics and macrophysics for each cloud element. We use cloud macrophysical properties (e.g., cloud size, cloud top bumpiness, CTH) and solar zenith angle to relate to 3-D radiative effects and test the sensitivity of the *Re* bias to these factors (Sect. 4.2). We also looked at the impact of drizzle on *Re* bias and found no apparent relationship between the two (details in Sect. 4.2.2).

**4.1 RSP cloud element analysis**

The cloud element labelling technique was developed using the RSP CTH retrievals. A contiguous set of CTH retrievals are counted as one cloud element. Places with no-retrievals between the cloud elements are labeled as "clear" segments. For each cloud element, means and standard deviations of cloud properties (*Re*, optical depth, CTH) are calculated along with the cloud elements' horizontal length. This method allows one to further relate the RSP retrieved cloud properties to quantities such as the standard deviation of CTHs and cloud horizontal length, which can serve as proxies for cloud top bumpiness and cloud size to further investigate the sensitivity of *Re* retrievals to these factors. When developing the cloud element labelling technique, we compared the cloud elements derived using RSP CTHs with that derived from HSRL-2 CTHs (Fig. 2(c)). While the two CTHs showed very similar results, we chose to use RSP since *Re* and COT are tied directly to RSP sampling.

Using all the cloud elements from the selected segments listed in Table 2, Fig. 13 shows mean *Re* values for each cloud element as a function of its mean CTH, with whiskers representing the standard deviations of *Re* and CTH for the cloud element. Each cloud element is color-coded by its cloud transect length. A prominent feature in Fig. 13 is the much-improved correlation between RSP polarimetric *Re* and CTH means, with a linear correlation coefficient of 0.72, compared to the correlation coefficient between RSP bi-spectral *Re* and CTH means of 0.24. The variability of bi-spectral *Re* is also much larger than that of polarimetric *Re* across all CTH levels, particularly for the lower levels (below 2 km). Colors also indicate that lower clouds are more often found with smaller transect lengths (less than 5 km, and often less than 1 km), indicating that low clouds are mostly very small cumuli. For clouds below 2 km, the largest differences between the two *Re* retrievals were as large as 20 μm. The mean difference between bi-spectral and polarimetric mean cloud-element *Re* across the four cases in Fig. 13 is 7.1 μm.





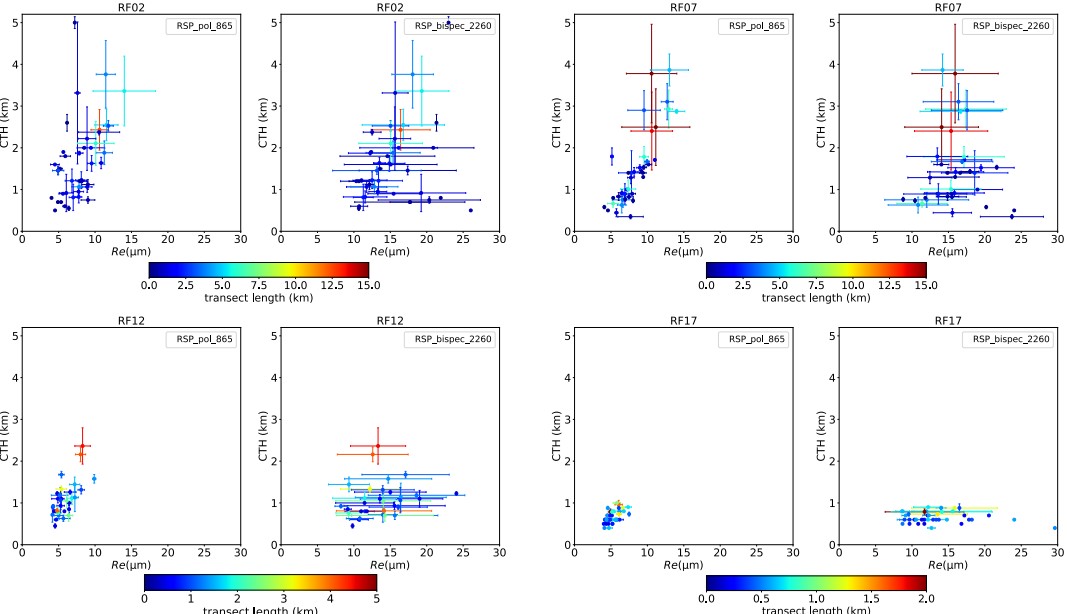

**Figure 13.** RSP Cloud element mean *Re* vs. mean CTH for the flight segments used in the four cross comparison cases in Sect. 3.3. The dots represent the mean *Re* value vs. mean CTH of each cloud element, the whiskers indicate standard deviations of *Re* and CTH for each cloud element. Cloud elements are color-coded by their horizontal transect lengths.

The same cloud element analysis was implemented for all research flights with good cloud sampling segments (number of cloud elements > 3), without further consideration for MODIS or in situ collocations. Thus, more RSP data is included when compared to the 4 individual cases. Table 4 lists the time periods of the 12 research flights used in Fig. 14. The findings are consistent with those for the 4 case studies above. Most importantly, the bi-spectral *Re* values are much larger than the polarimetric *Re* values across all CTH ranges. Across all 12 cases, the mean *Re* difference between the bi-spectral and the polarimetric *Re* is 11 μm. Figure 14 also shows that shallow cumulus clouds with CTH < 2 km usually have smaller sizes (horizontal length < 6 km), whereas the higher cloud tops are generally associated with larger CTH standard deviations, indicating bumpier cloud tops.

**Table 4.** The time periods from 12 Research Flights used to construct the analysis of Fig. 14.

| P-3 Flight Date by UTC start | RSP Time period (UTC) |
|---|---|
| 8/27/2019 RF02 | 8/27/19 3:30-4:30 |
| 8/30/2019 RF04 | 8/31/19 2:10-3:30 |
| 9/8/2019 RF07 | 9/9/19 1:00-2:12 |
| 9/13/2019 RF08 | 9/14/19 0:10-1:00 |
| 9/15/2019 RF09 | 9/16/19 3:00-4:00 |
| 9/16/2019 RF10 | 9/16/19 23:30-25:00 |
| 9/21/2019 RF12 | 9/22/19 2:12-5:40 |
| 9/23/2019 RF13 | 9/24/19 1:00-3:50 |
| 9/25/2019 RF14 | 9/26/19 3:30-5:30 |
| 9/27/2019 RF15 | 9/28/19 1:50-3:50 |
| 10/1/2019 RF17 | 10/2/19 1:50-4:30 |
| 10/5/2019 RF19 | 10/5/19 2:00-4:30 |



**Figure 14.** Same as Fig. 13 but derived from 12 research flights with good RSP sampling of warm cumulus and congestus clouds.



### 4.2 Sensitivity of *Re* retrieval bias to potential factors

In this section, the RSP cloud element statistics throughout the entire CAMP$^2$Ex mission are used to investigate the impact of various potential factors to the observed *Re* differences between RSP polarimetric and bi-spectral retrievals. The factors investigated include COT, cloud size, cloud top bumpiness, sub-pixel heterogeneity, SZA, and drizzle. In reality, the impacts of these factors are often intertwined. For example, clouds with smaller lengths are also shallower and optically thinner. Deeper clouds may also have larger lengths and increased likelihood of drizzle. While

it is not possible to fully isolate the impact from each individual factor, we will focus our discussion primarily within the scope of 3-D radiative effects, sub-pixel heterogeneity and drizzle.

### 4.2.1 3-D radiative effects and sub-pixel heterogeneity

*Cloud optical thickness and cloud size*

     Figure 15(a) shows the differences between the cloud element mean polarimetric *Re* and bi-spectral *Re*

retrievals organized as a function of cloud optical thickness. A sharp decrease of *Re* differences (bi-spectral − polarimetric) from ~25 μm to ~8 μm (maximum *Re* difference) is observed with increasing COT up to 9. This difference tapers off to a value of roughly 4 μm for larger COT. This pattern is similar to Fig. 5(a) in Marshak et al. (2006), where they used 3-D LES simulations to discuss the radiative effects of cloud's 3-D structure in 1-D *Re* retrievals from the bi-spectral technique. Marshak et al. (2006) concluded that in cumulus cloud fields, shadowing

effects (defined as when the measured reflectance is lower than its 1-D plane parallel equivalent reflectance) dominates over illumination effects (measured reflectance is higher than its 1-D plane parallel equivalent reflectance), and lead to an overestimate in retrieved 1-D *Re*. Vant-Hull et al. (2007) studied the impact of scattering angle on cumulus clouds to find that far from the backscatter direction (where shadowing effects dominate), 3-D cloud structures would lead to overestimates in the 1-D bi-spectral *Re* retrievals. However, near the backscatter viewing geometry (where

illumination effects dominate), the opposite is true. These effects in cumuliform cloud fields so far have been only studied through simulations. Since RSP bi-spectral retrievals were never taken close to the backscatter direction, the expectation is an overall overestimate of retrieved *Re* from the bi-spectral technique. If we assume the RSP polarimetric *Re* as the true *Re* (given how well it compared with in situ derived *Re* in Sect. 3.3), Fig. 15(a) suggests that in broken cumulus scenes (typical in CAMP$^2$Ex) large overestimates in bi-spectral *Re* dominate, being consistent

with these earlier studies that were based on simulations. But differences do exist. For example, we observe fewer negative values of scattered *Re* differences in Fig. 15(a) compared to Fig. 5(a) of Marshak et al. (2006). This may be due to different sun-view geometries, cloud structures, and cloud microphysics between RSP retrievals in CAMP$^2$Ex and the simulations in Marshak et al. (2006).

     When the *Re* and COT retrievals are further related to the cloud element's transect length as measured by RSP

(Fig. 15(b) and 15(c)), cloud elements with the largest *Re* differences (up to 25 μm) tend to be associated with the smallest transect lengths (< 1 km). As the transect length increases to 3 - 5 km, the large *Re* differences drop rapidly, showing no further dependence on transect length. Since smaller transect lengths are strongly associated with smaller clouds (with mean cloud area-equivalent diameters ~1.1 times larger than the mean of random linear transects though fields of cumuli, as shown by Barron et al. 2020), Fig. 15(b) reveals that clouds with the smallest sizes have the largest



*Re* differences between the two techniques. These smaller clouds are associated with smaller retrieved COT (Fig.
15(c)). These 1-D retrieved COT may be biased low by 3-D effects, such as shadowing and leakage of photons out
the side of clouds (e.g., Marshak et al. 2006). From Fig. 15, it is safe to conclude that RSP cloud element analysis
reveals that the largest *Re* differences are associated with clouds that are optically thin and small in horizontal size.

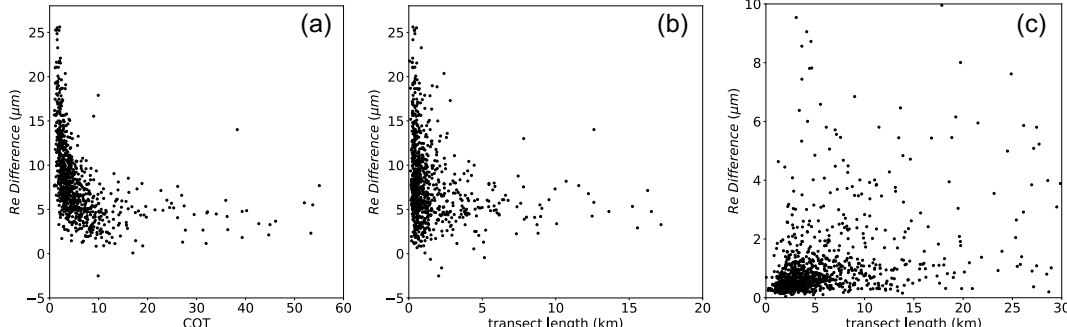

**Figure 15**. (a) *Re* difference (bi-spectral – polarimetric *Re*) vs. COT. (b) *Re* difference vs. transect length (c) Transect
length vs. COT. Each point represents an RSP cloud element.

*Clear sky contamination*

Another potential factor that could lead to considerable *Re* bias, as have been pointed out by several studies, is
sub-pixel reflectance variations (e.g., Marshak et al. 2006; Zhang et al. 2012; Zhang et al. 2016; Werner et al. 2018).
To give one such example, when partially cloudy regions are mixed with clear sky regions, low surface albedo (e.g.,
over ocean) would contribute to smaller nadir reflectance in the SWIR channel and lead to overestimated *Re* retrievals
compared to the true *Re* (e.g., Marshak et al. 2006). Attempts have been made to understand the magnitude of clear
sky contamination on *Re* retrievals. For example, Werner et al. (2018) used hi-resolution (30 m) ASTER data to
improve the bi-spectral *Re* retrievals for partly cloudy observations and showed the overestimates in retrieved *Re* at
MODIS pixel scales can exceed 41% due to clear sky contamination. For polarimetric *Re* retrievals, since it is sensitive
to the angular shape of the supernumerary bow from polarized reflectances, the presence of clear sky does not strongly
distort the shape of the supernumerary bows from cloud scattering, so polarimetric *Re* are not significantly affected
by clear sky contamination or sub-pixel heterogeneity. (e.g., Miller et al. 2018; Shang et al. 2015).

For CAMP²Ex, given the statistics of the sampled clouds (Fig. 2), the abundance of small (transect length) and
optically thin clouds (Fig. 15) led to our investigation of clear sky contamination to the observed *Re* differences
between RSP bi-spectral and polarimetric retrievals. The RSP instrument's 14 mrad IFOV converts to a roughly ~120
m horizontal footprint for the cloud retrievals. We used HSRL-2 2Hz data (~75 m horizontal resolution) to derive the
cloud fraction (CF) for each RSP cloud element as follows: For RSP cloud elements that have HSRL-2 CTH retrievals,
the HSRL-2 cloud fraction for a RSP cloud element is defined as the number of valid HSRL-2 CTH retrievals divided
by the total number of HSRL-2 CTH retrievals within the RSP cloud element. We examined the CDF of RSP cloud
elements against HSRL-2 CF and found the following: For clouds with transect lengths under 600m, at least 49% of
cloud elements have CF <0.95; for all cloud transect lengths, at least 46% of cloud elements have CF < 0.95. This
reveals that at least about half of the cloud elements have some degree of clear sky contamination (CF<0.95). We are



interested in how the amount of clear sky contamination can impact the observed differences between bi-spectral and
polarimetric *Re* retrievals.

To further investigate how *Re* differences depend on HSRL-2 CF, the cloud elements are separated into partly cloudy (CF < 0.5), mostly cloudy (0.5 < CF < 0.95) and overcast (CF > 0.95), and further divided into small transect length (< 2 km) and large transect length (> 2 km) populations. Figure 16 shows the PDFs of the three CF groups segregated by the two cloud size groups. For transect lengths smaller than 2 km (solid lines in Fig. 16), the three CF groups have similar distributions, with median values of *Re* differences of 8.9 μm, 7.3 μm and 8.1 μm for partly cloudy, mostly cloudy and overcast, respectively. For transect lengths greater than 2 km (dash lines in Fig. 16), there are no samples for CF < 0.5 (hence no red dashed histogram) and the median values of the *Re* differences are 5.3 μm and 5.7 μm for mostly cloudy and overcast groups, respectively. Thus, sub-pixel clear-sky contamination (derived from HSRL-2 2Hz) appears to account for < ~1 μm in the *Re* differences, with transect lengths playing a larger role since the median value of the small-size group being 2.4 μm larger than the large-size group in the *Re* differences. To summarize, from the RSP cloud element analysis, the large *Re* difference between the two techniques is found to be more related to optical depth and transect length than sub-pixel clear-sky contamination for the RSP warm cloud samples observed during CAMP²Ex.

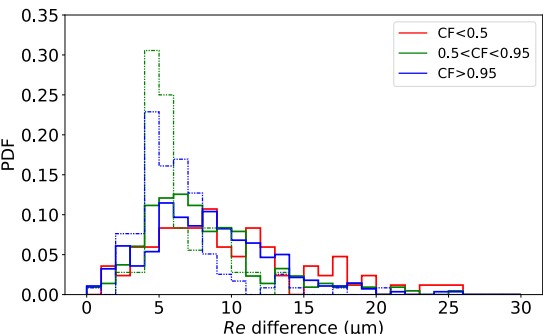

**Figure 16.** PDF of *Re* difference stratified by CF < 0.5 (11% of all cloud elements), 0.5 < CF < 0.95 (34% of all cloud elements) and CF > 0.95 (54% of all cloud elements) for cloud elements with transect lengths less than 2 km (79% of all cloud elements, solid) and greater than 2 km (21% of all cloud elements, dash).

*Cloud top bumpiness and solar zenith angle*

In our cloud element analysis, we defined cloud top bumpiness as the standard deviation of CTH for each cloud element, with the idea that it can capture the variation in cloud top structure for each cloud element. As pointed out by Loeb et al. (1998), not accounting for sub-pixel variations in CTH in the plane parallel assumption leads to large biases in retrieved COT, particularly at large SZA. More recent study has shown that polarimetric *Re* retrievals are less susceptible to cloud top bumpiness (e.g., Cornet et al. 2018). In our analysis, we used HSRL-2 2 Hz CTH to derive cloud top bumpiness, where the 2 Hz HSRL-2 data converts to a roughly ~50 m horizontal resolution. To avoid clear sky contamination, we separated overcast cloud elements from partly and mostly cloudy cloud elements as above. When the differences in RSP bi-spectral and polarimetric *Re* retrievals are organized as a function of HSRL-2 cloud top bumpiness, no apparent dependence was observed in the overcast cases (not shown). This might be due to the fact





that the possible effects of cloud top bumpiness on *Re* differences is masked by the effect of cloud size, since the clouds with the smallest transect length also have the smallest CTH standard deviation.

Past literature that examined the dependence of *Re* bias on SZA through simulations and observation studies typically show that SZA contributions to *Re* variations of ~1 to 2 μm. (e.g., Zhang et al. 2012; Grosvenor and Wood 2014; Horváth et al. 2014; Ahn et al. 2018). We also examined the impact of Solar Zenith Angle (SZA) on the RSP *Re* retrievals with our cloud element analysis, noting that the RSP retrievals rarely had cloud samples under low sun conditions (SZA <60°), with most samples taken with SZA between 20° and 45°. Similar to the findings in Ahn et al.

(2018) and Grosvenor and Wood (2014), the cloud element *Re* difference does not seem to be sensitive to SZA, possibly due to the small range of SZA in which RSP retrievals were collected, along with the complexity of the co-variability between different cloud variables and 3D pathways (e.g., shadowing vs illumination, leakage vs channeling).

### 4.2.2 Drizzle

In our analysis, we used the maximum APR-3 W-band reflectivity over a cloud element as a proxy for in-cloud drizzle for that cloud element. We examined the sensitivity of the differences between RSP bi-spectral and polarimetric retrievals of *Re* to the APR-3 reflectivity. Past studies on drizzle identification from W-band reflectivity showed that a threshold ~0 dBZ is associated with high likelihood of drizzle (e.g., Dzambo et al. 2019; Wang and Geerts 2003; Sauvageot and Omar, 1987). Using the cloud elements resulting from the segments included in the four cross-

comparison cases in Sect. 3.3 as an example, we applied a threshold of APR-3 W band maximum reflectivity of 0 dBZ to each cloud element to separate drizzle and non-drizzle cloud elements (Fig. 17). The maximum reflectivity is defined as the maximum of APR-3 W band column maximum reflectivity for each cloud element. After applying this drizzle filter, our results show that non-drizzling clouds typically exist for clouds with cloud tops below 3 km (Figs. 17(a) and 17(b)), yet drizzle clouds can exist at any level throughout the CTHs (Figs. 17(c) and 17(d)). For these non-

drizzling cloud elements, the *Re* differences between RSP polarimetric and bi-spectral retrievals still persist. In fact, when calculated, the mean *Re* difference for the non-drizzling cloud elements is 7.6 μm, and 4.9 μm for drizzling cloud elements. Separating the cloud elements by APR-3 drizzle detection simply did not show clear distinction in the mean *Re* differences.

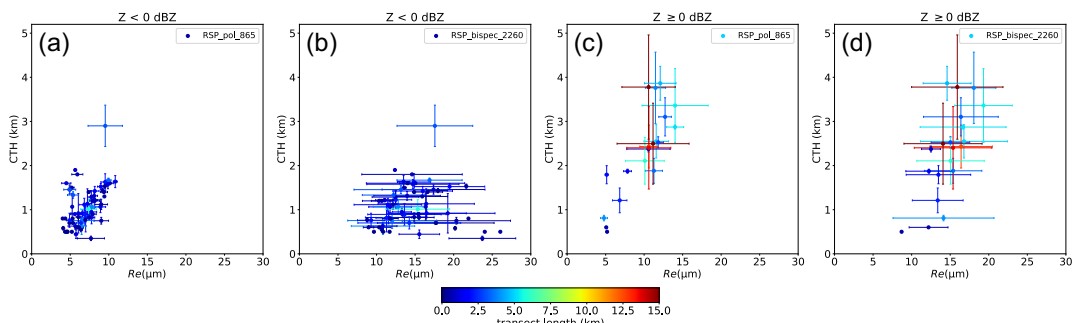

**Figure 17.** All the cloud elements from the 4 individual case studies, separated by APR-3 W-band max reflectivity Z = 0 dBZ: (a) RSP polarimetric *Re* with Z < 0 dBZ; (b) RSP bi-spectral *Re* with Z < 0 dBZ; (c) RSP polarimetric *Re*





with Z > 0 dBZ; (d) RSP bi-spectral *Re* with Z > 0 dBZ. Dot indicates the mean *Re* and mean CTH, while the horizontal and vertical whisker bars show the standard deviation of *Re* and CTH. All cloud elements are color-coded by its horizontal length.

We further examined all P-3 research flights and binned all RSP cloud elements by APR-3 W band maximum reflectivity, separating drizzle and non-drizzle cloud elements using Z = 0 dBZ threshold. The mean and median *Re* bias for non-drizzling cloud elements is 8.6 µm and 7.5 µm, respectively; for drizzling cloud elements, they are 5.8 µm and 4.9 µm. To avoid the possible correlation between COT and radar reflectivity, we also tested using only cloud elements with COT > 10, and we did not find a different trend. This is similar to the results from using the segments

from the 4 case studies (Fig. 17), both showing larger *Re* difference for non-drizzling cloud elements. When all the cloud elements are binned by W-band reflectivity in 5 dBZ intervals (Fig. 18), the largest mean *Re* difference of 10.1 µm (with a standard deviation of 5.8 µm) was observed for 0 dBZ < Z < 5 dBZ, and the smallest mean *Re* difference of 3.5 µm (with a standard deviation of 2.0 µm) was observed for 15 < Z < 20 dBZ. Figure 18 also reveals a large range of *Re* difference values (of up to 25 µm) for bins with Z < 0 dBZ, whereas a trend of decreasing range and

maximum *Re* difference is observed for bins with Z > 0 dBZ. While no clear sensitivity of *Re* difference on W-band Reflectivity is suggested, Fig. 18 indicates that observed large *Re* differences especially for bins with Z < 0 dBZ could not be explained by drizzle. Therefore, we conclude that it is not likely that the difference in the bi-spectral and polarimetric *Re* can be explained by drizzle. This aligns with findings from Zhang et al. (2012) and Ahn et al. (2018).

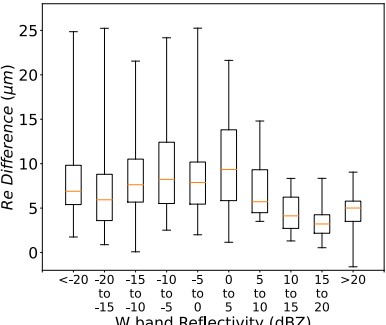

**Figure 18**. RSP cloud element *Re* difference using all RSP cloud elements across all RFs, binned by APR-3 W-band column Max reflectivity intervals of 5 dBZ. Orange line indicates median for each bin, the box ends indicate interquartile, and the end of whiskers indicate maximum and minimum values for each bin.

**4.3 Consistency of *Re* retrieval representativeness from CAMP²Ex**

Lastly, the consistency between *Re* retrievals across all techniques is examined for the CAMP²Ex region. In doing

so, we seek to gauge the representativeness of Terra-MODIS *Re* retrievals by comparing to RSP (airborne remote sensing) and in situ measurements sampled across all RFs. Level 2 liquid *Re* retrievals from Terra-MODIS within the CAMP²Ex region (Fig. 1) from all 19 P-3 Research Flight days were included to derive a *Re* distribution. All valid *Re* retrievals from RSP and MODIS *Re* are included after removing cirrus and ice clouds indicated by SPN-S and MODIS phase flag. For in situ measurements, again to avoid sampling differences with passive remote sensing (i.e.,

in situ sampling through deep convective clouds which was not sampled by RSP), in situ samples were removed with 11µm Brightness Temperature < 273K as indicated from AHI. Figure 19 shows the *Re* distributions from MODIS and





RSP and in situ measurements. Given the difference in resolution and spectral channel, RSP-polarimetric and bias-adjusted MODIS $Re$ agrees within 1 μm as indicated by the median and mean values in Table 5. The two also have very similar variability as indicated by the standard deviations. On the other hand, RSP bi-spectral and the original

MODIS $Re$ also have similar $Re$ statistics that agree within ~2 μm, but both are also 5-7 μm larger than RSP-Polarimetric and bias-adjusted MODIS Re with larger standard deviations. In situ derived $Re$ from the P-3 and Learjet indicates a median of 11.0 μm and 12.4 μm, which agrees to RSP-polarimetric and bias-adjusted MODIS $Re$ within ~ 2 μm. Longer tails in the distributions of $Re$ from in situ measurements are not limited by the 30 μm cut-off in the bi-spectral retrievals LUT. Also as noted in Sect. 3, are associated with aircraft penetration of deeper clouds not near the

same CTH level that contain drizzle. This long tail contributes to the much larger mean and standard deviation statistics relative to the remote sensing retrievals. Overall, RSP polarimetric $Re$ and bias-adjusted MODIS $Re$, the Learjet, P-3 in situ indicated similar median $Re$ values of ~10 - 12 μm, while MODIS $Re$ and RSP bi-spectral $Re$ show overestimates of ~ 5 - 7 μm compared to the other techniques. This is also consistent with the results from the individual case studies in Sect. 3.3.

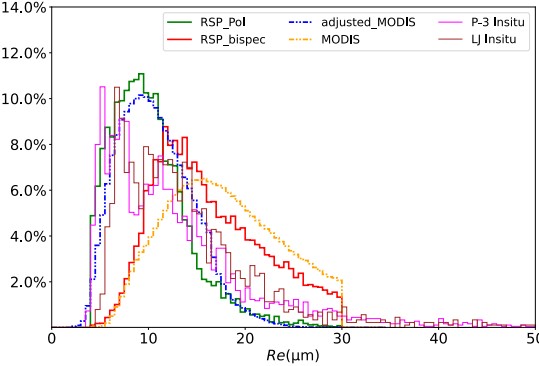


**Figure 19.** $Re$ distributions from L2 RSP polarimetric $Re$, L2 RSP bi-spectral $Re$, L2 MODIS $Re$, L2 MODIS bias-adjusted $Re$, 1 Hz P-3 in situ derived $Re$, 1 Hz Learjet in situ derived $Re$ using all valid retrievals of warm oceanic clouds within the domain indicated in Fig. 1.

**Table 5.** $Re$ statistics averaged over all research flight segments for warm clouds.

| | Median (μm) | Mode (μm) | Mean ± Standard deviation (μm) |
|---|---|---|---|
| **RSP Polarimetric $Re$** | 9.6 | 9.0 | 10.2 ± 4.0 |
| **MODIS Adjusted $Re$** | 10.4 | 9.0 | 10.8 ± 3.8 |
| **P-3 in situ $Re$** | 11.0 | 5.0 | 13.6 ± 11.3 |
| **Learjet in situ $Re$** | 12.4 | 6.5 | 15.2 ± 12.1 |
| **RSP bi-spectral $Re$** | 15.1 | 11.5 | 16.2 ± 5.5 |
| **MODIS $Re$** | 17.2 | 15.5 | 17.7 ± 5.7 |

To test the similarity of the different $Re$ distributions shown in Fig. 19, a Kolmogorov-Smirnov (K-S) test was applied to all 6 $Re$ distributions, with the results listed in Table 6. When the other 5 $Re$ distributions are compared with the RSP polarimetric $Re$ distribution, both the RSP bi-spectral $Re$ and MODIS $Re$ distributions have p-values < 0.05, which indicates that these two $Re$ distributions do not belong to the same distribution as the RSP polarimetric $Re$ distribution. Bias-adjusted MODIS, P-3 in situ and Learjet in situ $Re$ distributions, however, have p-values > 0.05,

and therefore we cannot reject the null hypothesis that these three $Re$ distributions belong to the same distribution as





the RSP polarimetric *Re* distribution. Among the three distributions, bias-adjusted MODIS *Re* has the smallest K-S value and the highest p-value, which indicates it is overall the closest fit to RSP polarimetric *Re* distribution. The K-S results are consistent with the difference in the *Re* median values from Table 5, i.e., RSP polarimetric, bias-adjusted *Re*, P-3 in situ and Learjet in situ have closer median *Re* values within their differences within 1 to 2 μm, while RSP

bi-spectral *Re* and MODIS Re median values are 5 to 7 μm larger than that from the RSP-polarimetric *Re*. We also examined the similarity for CTH distributions retrieved from RSP and MODIS, and obtained a K-S statistic of 0.32 with a p-value of 0.31. Therefore, we cannot reject the null hypothesis that the two CTH distributions belong to the same distribution. Similarly, when Learjet and P-3 sampling altitude distributions averaged across all flight segments are compared, the two in situ sampling distributions have a K-S statistic of 0.16 and a p-value of 0.98. Thus, we

conclude that the two in situ techniques sampled similar cloud fields during CAMP²Ex, and RSP and MODIS cloud samples also came from similar cloud fields during CAMP²Ex.

**Table 6.** Kolmogorov-Smirnov test for similarity statistics, using RSP Polarimetric *Re* distribution as the reference.

| Compared to RSP Polarimetric *Re* dist. | K-S statistics | p-value |
|---|---|---|
| RSP Bi-spectral | 0.31 | 0.008 |
| MODIS | 0.34 | 0.002 |
| Bias-adjusted MODIS | 0.10 | 0.92 |
| P-3 in situ | 0.20 | 0.09 |
| Learjet in situ | 0.19 | 0.112 |

## 5 Conclusions

This paper presents the first field evaluation of satellite bi-spectral *Re* retrievals in tropical cumulus cloud fields.

The evaluation consists of comparison between airborne RSP bi-spectral and polarimetric retrievals of *Re*, and cross-comparison between airborne remote sensing, in situ and satellite retrieved *Re* collected during the CAMP²Ex field campaign. Unlike previous studies that used field data for evaluating satellite bi-spectral retrievals of stratocumulus cloud fields, validation in cumulus cloud fields presents a greater challenge since they are less persistent, with fast changing cloud morphologies and complex cloud structures. Here, we take a full advantage of RSP's capability to

provide both collocated bi-spectral and polarimetric *Re* retrievals; thus there is no sampling difference in comparing the two retrieval techniques. We show that the RSP bi-spectral *Re* retrieved in CAMP²Ex cloud fields is on average overestimated by 6.0 μm compared to the RSP polarimetric *Re* for the 1.2 Hz samples across the entire mission. RSP polarimetric *Re* also indicates much less variability and a clear increase with CTH compared to RSP bi-spectral *Re*.

MODIS *Re* retrievals, which uses a bi-spectral approach, is in good agreement with RSP bi-spectral *Re* (median

*Re* difference within 2.1 μm from Table 5). The bias-adjusted MODIS *Re*, based on the Fu et al. (2019) bias-correction factors, shows tight agreement with the RSP polarimetric *Re* (median *Re* difference within 0.8 μm from Table 5). The bias-adjusted MODIS *Re* and the RSP polarimetric *Re* both show increasing profiles with CTH, and less variability compared to the original MODIS *Re*. The in situ measured *Re* values are in good agreement (median *Re* difference within 2.8 μm from Table 5) with the RSP-polarimetric and bias-adjusted MODIS *Re* values. Further restricting

altitudes to CTH < 2 km for shallow convection yields better agreement (median *Re* within 1.7 μm) between them.


Thus, these three independent techniques are in very good agreement with each other and are ~5.5 – 7.6 μm smaller than the median $Re$ values from the bi-spectral $Re$ from RSP and MODIS. These agreements were found to be consistent between mission averaged statistics (Table 5) and case by case comparison (Table 3). For deeper clouds containing in situ measured precipitation, in situ measures of $Re$ can at times be much larger than all of the remotely sensed $Re$ values.

By taking advantage of collocated RSP, APR-3 and HSRL-2 on the P-3, we further examined the differences in RSP bi-spectral and polarimetric $Re$ and how they relate to cloud macrophysics (cloud transect length, cloud top bumpiness, sub-pixel cloud fraction), COT, SZA and drizzle. We found that $Re$ differences (bi-spectral – polarimetric) of up to 25 μm (median 11 μm) is associated with small COT (COT < 5). As COT increases from 5 to 15, the $Re$ difference maximum decreases to ~ 5 μm (median ~3 μm). For COT greater than 15, there is no clear dependence of $Re$ difference on COT. Similarly, $Re$ differences of up to 26 μm (median ~8 μm) are associated with the smallest cloud transect lengths (< 0.5 km). For cloud transect lengths greater than 5 km, $Re$ differences drop to 10 μm (median ~5 μm). RSP cloud retrievals have clear sky contamination, as revealed by higher resolution HSRL-2 data. Clear sky contamination is shown to have only a minor impact on $Re$ differences (< 1μm) relative to fully cloudy pixels. No apparent relationships between $Re$ differences and SZA and cloud top bumpiness are observed, noting that the range of SZA sampled during RSP $Re$ retrievals was small under moderately high sun condition (SZA = 20° to 45°) and the co-variability of cloud top bumpiness with other cloud variables. A third of the cloud elements sampled by RSP contained drizzle as revealed by APR-3. No apparent relationship between $Re$ differences and maximum W-band reflectivity are observed. On average, cloud elements with detectable drizzle have $Re$ differences that are ~ 1 μm smaller than cloud elements with no detectable drizzle.

Our analysis in Sect. 3.1 showed that most samples observed by the P-3 remote sensors came from small, optically thin, non-drizzling, shallow clouds. The samples exhibit a large difference (~factor of 2) between RSP bi-spectral and polarimetric $Re$ retrievals. For non-drizzling shallow clouds, in situ observations compare well against the RSP polarimetric retrievals, and show a vertical variability of a few microns. For these non-drizzling shallow clouds, no in situ $Re$ samples are as large as the RSP bi-spectral $Re$. Therefore, for the shallow clouds observed by RSP during CAMP²Ex, the long-held hypothesis of the presence of drizzle or vertical variations as major contributing factors to $Re$ differences between bi-spectral and polarimetric retrievals could be rejected with near certainty. Thus, for the shallow, non-drizzling clouds, the evidence presented herein is strongly suggestive that the dominant cause for the differences between RSP polarimetric and bi-spectral $Re$ observed during CAMP²Ex lies within 3D radiative pathways that lead to large positive biases in bi-spectral retrievals of $Re$ compared to polarimetric retrievals. For deeper clouds that contain drizzle, true in-cloud vertical variations could still be at play in explaining additional $Re$ differences between bi-spectral and polarimetric techniques.

For MODIS, there is a substantial number of partly cloudy pixels as revealed by coincident, high-resolution ASTER data. These sub-pixel clouds often lead to failed MODIS retrievals of $Re$, as discussed in Cho et al. (2015). Comparing the cloud macrophysical properties for CAMP²Ex reported in Sect. 3.1 with those reported for RICO and INDOEX, the CAMP²Ex shallow clouds are much smaller. We speculate that the reason for the maximum failure rate in MODIS cloud microphysical retrievals occurring over the western tropical Pacific, as reported by Cho et al. (2015),



may be because of the high frequency of small clouds here relative to anywhere else. Still, as shown in Sect. 4.3, we cannot reject the null hypothesis that the MODIS CTH distributions belong to the same distributions observed by the
RSP from P-3. This provides confidence that the conclusions drawn from the RSP polarimetric and bi-spectral *Re* comparison extend to MODIS as well.

This study also provides additional validation of the bias-adjusted MODIS *Re* values reported in Fu et al. (2019), showing a mission averaged mean ± standard deviation of 10.8 ± 3.8 μm compared to RSP polarimetric *Re* values of 10.2 ± 4 μm as shown in Table 5. Throughout, we used the upper-bound *Re* bias adjustment factors of Fu et al. (2019).
Using the lower-bound bias adjustment factors leads to a mission mean ± standard deviation of 6.7 ± 3.2 μm. The RSP polarimetric *Re* falls within these bounds. Fu et al. (2019) showed that the largest regional *Re* biases for marine liquid water clouds occur over the tropical western pacific, and our results seem to indicate that this may be because of a higher frequency of smaller clouds here relative to everywhere else. Our validation here, along with in situ validation of MODIS *Re* from other regions (e.g., Painemal and Zuidema 2011, Ahn et al. 2018), provides additional confidence
in the global distribution of bias-adjusted MODIS *Re* reported in Fu et al. (2019).

*Data availability*. CAMP²Ex datasets of RSP, APR-3, HSRL-2, SPN-S, P-3 collocated AHI CLAVR-X data products, SPEC in situ data used in this analysis are available at: https://www-air.larc.nasa.gov/cgi-bin/ArcView/camp2ex. CAMP²Ex P-3 and Learjet forward videos can be access at: https://asp-archive.arc.nasa.gov/CAMP2EX/N426NA/video/ and https://www-air.larc.nasa.gov/cgi-
bin/ArcView/camp2ex?LEARJET=1. MODIS Collection 6.1 cloud products (dx.doi.org/10.5067/MODIS/MOD06_L2.061), MODIS Level 1B Calibrated radiances at 250m (dx.doi.org/10.5067/MODIS/MOD02QKM.061) and 500m (dx.doi.org/10.5067/MODIS/MOD02HKM.061) were obtained through the level 1 and Atmosphere Archive and Distribution System of NASA Goddard Space Flight Center (https://ladsweb.modaps.eosdis.nasa.gov). The MODIS bias-adjusted *Re* correction factors can be found at:
https://doi.org/10.17632/j4r72zxc6g.2. AHI standard cloud products are available at: https://www.eorc.jaxa.jp/ptree/index.html. The 10-minute AHI 1km reflectances imagery can be accessed from the CAMP²Ex Worldview interface (http://geoworldview.ssec.wisc.edu).

*Author contribution*. DF performed the analysis and drafted the manuscript. The methodology was developed by DF, LDG. LDG, JL, BvD, YH, GMM, RMR, SWN helped with the writing and editing of the manuscript at various stages.
RSP retrievals used in this study was supported by BC, MA, BvD. PL, SW collected, processed, and curated in-situ measurements used in this study. APR-3 data was supported by TS, OOS collected, processed, and curated APR-3 data. CH and AJS collected, processed, and curated HSRL-2 dataset. SS provided the SPN-S dataset and YH derived SPN-S transmittance. All authors contributed to the editing of the manuscript.

*Acknowledgements. The authors would like to acknowledge NASA grant numbers 80NSSC18K0144,*
*80NSSC18K0150, 80NSSC18K0146, and 80NSSC21K1449 and their program manager, Dr. Hal Maring. We thank all the members of the CAMP²Ex team for their hard work in collecting the datasets analyzed herein. We also thank Dr. Guangyu Zhao for his help in providing reprojected overlays of ASTER and MODIS data.*



*Competing interests.* The authors declare that they have no conflict of interest.

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
