# Peer review of "An evaluation of liquid cloud droplet effective radius derived from MODIS, airborne remote sensing and in situ measurements from CAMP2Ex"

_Atmospheric Chemistry and Physics, 2022_

## Referee Comment (RC1)

**Review of Fu, et al., 2022.**

This paper shows results from an aircraft field campaign in the Philippines where mostly cumulus – like clouds were sampled and focuses on the retrieval of cloud droplet effective radius (Re) using remote sensing. The aircraft were equipped with both the remote sensing RSP polarimeter instrument and in-situ probes. The latter can directly measure cloud droplet size distributions (and hence Re). The RSP instrument can retrieve Re using both polarimetry and using the bi-spectral method. The latter is similar to the method used by the MODIS satellite instrument. The polarimetry method is thought to be less prone to retrieval errors than the bi-spectral method. The results show that the bi-spectral method produces higher Re values than the polarimetery method. The polarimetery method agrees better with the in-situ values except in some cases (where rain is blamed for increasing the in situ Re). MODIS retrievals that are approximately collocated produce similar values to the bi-spectral RSP method. A correction for MODIS is also applied and this brings the MODIS values close to the in-situ and polarimetery values. The bi-spectral errors are largest for small cloud optical depth values. Some potential reasons for the error using the bi-spectral technique are explored with the data suggesting that some are not likely explanations.

Overall, the paper shows catalogues some nice results. The main results are important and are worthy of publication; namely, that the bi-spectral aircraft RSP retrievals overestimate Re, that the MODIS bi-spectral results agree with the aircraft bi-spectral Re, that the MODIS correction seems to work well and that the bias is worse at low optical depths. The text is also well written and mostly clear, but the number of figures is a little excessive. As mentioned below I would like to see the analysis look more closely at the issue of sub-pixel variability – or at least to discuss it more thoroughly. I would be happy to see it published once the above issues are addressed.

The paper doesn't really get to the bottom of what causes the high values for the bi-spectral retrievals, except that they mainly occur at low optical depths. From previous work (e.g., see Fig. 3d) and e) of Zhang (2016) it seems clear that we might expect large Re retrieval biases at low optical depths due to the non-linear nature of the look-up tables (LUTs) used in the retrievals combined with sub-pixel variability of COD. At low COD values the LUTs get very non-linear making them more susceptible to these errors, and also they become more sensitive to reflectance errors. This should be discussed more in the paper as a possible explanation. Why not examine the effect of cloud heterogeneity on the Re biases using the variability in COT or in 0.86um reflectance (e.g., as used in Liang, 2009 and Zhang, 2016)?

Regarding clear-sky contamination (Fig. 16) using the HRSL-2 lidar – what is the optical depth threshold used to define regions as cloud when estimating the cloud fraction? Is it comparable to that used for the bi-spectral RSP or MODIS retrievals? Perhaps if the HRSL-2 is very sensitive to cloud then its definition of cloud fraction within a cloudy segment is not meaningful for bi-spectral retrievals. What happens if you use a higher optical depth threshold?

The ASTER data (Fig. 12) does not seem to be used except to show that many of the clouds are smaller than 1km in size. Could it be used to estimate the effects of sub-pixel cloud heterogeneity (similar to what was done in Werner, 2018)? Since this could be a cause of the Re biases. Also, can you examine what the MODIS PCL retrievals estimate for Re?

I don't really see any evidence for this statement in the abstract (line 44): "3D radiative pathways appear to be the leading cause for the large positive biases in bi-spectral retrievals.". Where was this shown?

Regarding the bias correction to the MODIS data – do the results here suggest that the lower bound of the correction is most suitable? Could this be added as a recommendation at the end?

Some of the figures seem a little bit redundant and the paper is very long in terms of the number of figures. Perhaps some of them could be removed, or put into an appendix or supplementary section? E.g., Figs. 13 and 14 don't seem to add much beyond what has already been shown. If you want to demonstrate that the CTH variability gets larger with height then there are more direct ways to do this. Fig. 15a seems the same as Fig. 3b and I'm not sure how much the other panels add for Fig. 15. The 4 figures for the different case studies start to get a bit repetitive too. Figures 6 and 7 are useful perhaps since Fig. 7 shows some interesting high Re values from the Lear Jet. But after that the story is similar with higher Re retrievals for the bi-spectral method. Maybe the rest of the case study figures (Figs. 8-11) could be put into an appendix?

**Specific comments**

Fig. 2 – the caption should say whether the data is for all of flights for which data was captured and for only oceanic liquid clouds only.

L357 – "The samples exhibited a large difference (a factor of ~2) between RSP bi-spectral and polarimetric Re retrievals, which is investigated further in the sections below."

- This sentence seems out of place at the end of a paragraph about radar reflectivity. I would argue that it is not necessary since it is clear by now that the paper is focuses on Re. But if you want to keep this to lead into the focus on Re then it would be better in its own paragraph and would need some rewording.

"Cloud bow COT" – this is used a few times and is only explained in the Results section rather than in the Methods – a short description of how this is done would be better placed in Section 2.1.1 then in the Results section. I also think that just calling it the RSP COT would be clearer (after you have defined what this means). It would then be more aligned with how you describe the Re from the RSP.

Fig. 4h - it's a bit hard to see the CTH dots – would this be better as a separate timeseries and maybe with a line as well as dots? This could perhaps be overlaid on the radar reflectivity plot.

Fig.7 – it's interesting that here the RSP-pol and adjusted MODIS values agree better with the in-situ values at lower altitudes, but at higher altitudes the bispectral and non-adjusted MODIS values do. You should mention this even though you suggest that the Lear jet samples were biased high due to flying through a raining turret (i.e., sampling bias – this could be spelled out a bit more clearly in the text). Perhaps other explanations could be a potential role of cloud top entrainment causing the RSP-pol Re retrievals to be low (since they sample very close to cloud top), whereas deeper in the clouds (in-situ and bi-spec retrievals) the Re values are higher?

Figs. 4, 8 and 10 – the captions should say which flight the plots were were for (RFxxx).

Case study introductory paragraphs – i.e., Sections 3.3.1 - 3.3.4. These should mention the relevant overview figures early on (i.e., Figs. 4,6 8 and 10) when giving the general description of the cases.

Fig. 12 – what happens if you use the PCL retrievals from MODIS?

Figs. 13 and 14 – are these necessary? I'm not sure that they add much beyond what has already been shown. If you want to demonstrate that the CTH variability gets larger with height then there are more direct ways to do this.

Fig. 15a – how is this different to Fig. 3b? Do panels b and c add much more? And would they be better as density maps instead of scatter plots as in Fig. 3?

L743 – "to derive the cloud fraction (CF) for each RSP cloud element as follows: For RSP cloud elements that have HSRL-2 CTH retrievals, the HSRL-2 cloud fraction for a RSP cloud element is defined as the number of valid HSRL-2 CTH retrievals divided by the total number of HSRL-2 CTH retrievals within the RSP cloud element"

- This is a little confusing – do you mean that you divide by the total number of *attempted* HSRL-2 CTH retrievals?

L756 - "there are no samples for CF < 0.5 (hence no red dashed histogram)" - it would be good to mention this in the caption of the figure (16) too.

**Typos / grammar**

L41 – "RSP bi-spectral Re shows larger relative values compared to RSP polarimetric Re for smaller and optically thinner clouds." – this doesn't quite get across the result that the bias is worse for smaller optical depths. I recommend :- "The overestimate of Re from the RSP bi-spectral method relative to Re from the polarimetric RSP method increased as cloud size and optical depth reduced."

L69 – "(hence, 1-D radiative transfer as the forward model used in this retrieval)," – better as "(hence, 1-D radiative transfer is used as the forward model in this retrieval)," I think.

L142 – "because the overpass time of the latter two sensors occurs in the afternoon when cirrus is more frequent and when the aircraft was returning to base that did not have favourable samplings.". Better as :- "because the overpass time of the latter two sensors occurs in the afternoon when cirrus is more frequent and when the aircraft was returning to base; therefore, the sampling was not favourable."

L174 – "extend its evaluation in cumulus cloud fields" -> "extend its evaluation to cumulus cloud fields"

L179 – "Mean and standard deviations of retrieved quantities belonging to a cloud element is computed." -> "Means and standard deviations of retrieved quantities are computed for cloud elements."

L200 – "Only drop size distribution with" -> "Only drop size distributions with" + "temperature" -> "temperatures"

L211 – "The median differences within 1  $\mu$ m" – "The median differences were within 1  $\mu$ m".

L256 - "pixel" -> "pixels".

L268 – "to retrieve COT at MISR 9 view angles" -> "to retrieve COT at MISR's 9 view angles"

L267 – "As a continuation of Liang et al. (2015), fused MISR L1B radiance data and MODIS L2 cloud

Re were used to retrieve COT at MISR 9 view angles.".

Better as :- "As a continuation of Liang et al. (2015), Fu et al. (2019) fused MISR L1B radiance data and MODIS L2 cloud Re to retrieve COT at MISR's 9 view angles." (Since otherwise it sounds like this was done in the paper under review here).

L330 – "The median COT is 3.5 for the cloud bow (COT retrieved using total reflectances and polarimetric Re) and 4.2 for the bi-spectral." – add "retrievals" at the end.

L440 – "over which little overall change in the cloud field" – add "occurs" after this.

L441 – "Here, the Re retrievals were sorted into 250 m CTH bins." – this makes it sound like you are referring to a figure or a result that has already been introduced. You could just say "We sorted the Re retrievals into 250 m CTH bins."

L581 – "Apart from these clouds being sub-pixel to MODIS retrievals cannot resolve" -> "Apart from these clouds being sub-pixel to MODIS retrievals so that MODIS cannot resolve them"

L609 – "As mentioned in Sect. 2.2, we acknowledge the difference between the MODIS 2.1  $\mu$ m channel to the RSP 2.26 610  $\mu$ m channel, we acknowledge that the differences in the two wavelengths and we do not expect the Re retrieved from MODIS and RSP bi-spectral to have the exact same bias" -> – "As mentioned in Sect. 2.2, we acknowledge the difference between the MODIS 2.1  $\mu$ m channel to the RSP 2.26 610  $\mu$ m channel we acknowledge that the differences in the two wavelengths and so we do not expect the Re retrieved from MODIS and RSP bi-spectral to have the two wavelengths and so we do not expect the Re retrieved from MODIS and RSP bi-spectral to have the two wavelengths and so we do not expect the Re retrieved from MODIS and RSP bi-spectral to have the two wavelengths and so we do not expect the Re retrieved from MODIS and RSP bi-spectral to have the exact same bias"

L629 – "that can be" -> "than can be".

L660 – "A prominent feature in Fig. 13 is the much-improved correlation between RSP polarimetric Re and CTH means, with a linear correlation coefficient of 0.72, compared to the correlation coefficient between RSP bi-spectral Re and CTH means of 0.24." -> "A prominent feature in Fig. 13 is the much-improved correlation between the RSP polarimetric Re and CTH means (linear correlation coefficient, r, of 0.72) compared to between the RSP bi-spectral Re and CTH means (r=0.24)." L844 – "Also as noted in Sect. 3, are associated with aircraft penetration of deeper clouds not near

the same CTH level that contain drizzle." -> "Also as noted in Sect. 3, they are associated with the

penetration of deeper clouds at altitudes different to the CTH level observed by the remote sensing,

with the clouds tending to contain drizzle."

L846 – "Overall, RSP polarimetric Re and bias-adjusted MODIS Re, the Learjet, P-3

in situ indicated" -> "Overall, Re observations from the RSP polarimetric, the bias-adjusted MODIS,

the Learjet in situ and the P-3 in situ techniques indicated..."

L864 – "have closer median Re values within their differences within 1 to 2  $\mu$ m," – not clear what

you mean by "within their differences"?

**References**

Liang, L., Di Girolamo, L., & Platnick, S. (2009). View-angle consistency in reflectance, optical thickness and spherical albedo of marine water-clouds over the northeastern Pacific through MISR-MODIS fusion. Geophysical Research Letters, 36, L09811. https://doi.org/10.1029/2008GL037124

Zhang, Z., Werner, F., Cho, H. M., Wind, G., Platnick, S., Ackerman, A. S., et al. (2016). A framework based on 2-D Taylor expansion for quantifying the impacts of subpixel reflectance variance and covariance on cloud optical thickness and effective radius retrievals based on the bispectral method. Journal of Geophysical Research: Atmospheres, 121, 7007–7025. https://doi.org/10.1002/2016JD024837

Werner, F., Zhang, Z., Wind, G., Miller, D., & Platnick, S. (2018). Quantifying the impacts of subpixel reflectance variability on cloud optical thickness and effective radius retrievals based on high-resolution ASTER observations. Journal of Geophysical Research: Atmospheres. 123, 1–20. https://doi.org/10.1002/2017JD027916

---

## Author Comment (AC1)

*We would like to thank the editor for handling the review process for our manuscript. We would also like to thank both reviewers for their careful review of the manuscript. We have taken actions to modify the manuscript in response to the reviewers' suggestions. Our responses to their comments and questions are listed below. Comments from the reviewers are in upright plain texts and our replies are in italics. The line numbers in our responses are referring to the line numbers in the revised manuscript.*

**Reviewer 1:**

This paper shows results from an aircraft field campaign in the Philippines where mostly cumulus –like clouds were sampled and focuses on the retrieval of cloud droplet effective radius (Re) using remote sensing. The aircraft were equipped with both the remote sensing RSP polarimeter instrument and in-situ probes. The latter can directly measure cloud droplet size distributions (and hence Re). The RSP instrument can retrieve Re using both polarimetry and using the bi-spectral method. The latter is similar to the method used by the MODIS satellite instrument. The polarimetry method is thought to be less prone to retrieval errors than the bi-spectral method. The results show that the bi-spectral method produces higher Re values than the polarimetery method. The polarimetery method agrees better with the in-situ values except in some cases (where rain is blamed for increasing the in situ Re). MODIS retrievals that are approximately collocated produce similar values to the bi-spectral RSP method. A correction for MODIS is also applied and this brings the MODIS values close to the in-situ and polarimetery values. The bi-spectral errors are largest for small cloud optical depth values. Some potential reasons for the error using the bi-spectral technique are explored with the data suggesting that some are not likely explanations.

Overall, the paper shows catalogues some nice results. The main results are important and are worthy of publication; namely, that the bi-spectral aircraft RSP retrievals overestimate Re, that the MODIS bi-spectral results agree with the aircraft bi-spectral Re, that the MODIS correction seems to work well and that the bias is worse at low optical depths. The text is also well written and mostly clear, but the number of figures is a little excessive. As mentioned below I would like to see the analysis look more closely at the issue of sub-pixel variability – or at least to discuss it more thoroughly. I would be happy to see it published once the above issues are addressed.

**Response:** *We are glad the reviewer liked the paper. We address all of the reviewer's comments below.*

The paper doesn't really get to the bottom of what causes the high values for the bi-spectral retrievals, except that they mainly occur at low optical depths. From previous work (e.g., see Fig. 3d) and e) of Zhang (2016) it seems clear that we might expect large Re retrieval biases at low optical depths due to the non-linear nature of the look-up tables (LUTs) used in the retrievals combined with sub-pixel variability of COD. At low COD values the LUTs get very non-linear making them more susceptible to these errors, and also they become more sensitive to reflectance errors. This should be discussed more in the paper as a possible explanation. Why not examine the effect of cloud heterogeneity on the Re biases using the variability in COT or in 0.86um reflectance (e.g., as used in Liang, 2009 and Zhang, 2016)?

*Response: We did discuss the issues leading to the large Re biases for low cloud COD in the first few paragraphs of Section 4. This included issues with the LUT approach for low COD, radiometric calibration uncertainty, surface reflectance uncertainties, sub-pixel variability, etc., as a lead in to the extensive RSP analyses presented in the rest of Section 4. These analyses were synthesized in Section 5, concluding that:*

*Line 847-850: "… for the shallow, non-drizzling clouds, the evidence presented herein is strongly suggestive that the dominant cause for the differences between RSP polarimetric and bi-spectral Re observed during CAMP²Ex lies within 3-D radiative pathways that lead to large positive biases in bi-spectral retrievals of Re compared to polarimetric retrievals."*

*As for examining the Re bias using variability in COT or in 0.86 µm reflectance, we did do this before submitting the manuscript, but chose not to show the results. From the RSP dataset, we used the RSP 0.86um reflectance and applied a 3-pixel 1-d window to calculate the standard deviation over the mean reflectance as a measured cloud horizontal heterogeneity. Note that this just captures resolved variability. This is not the same as Hσ reported by MODIS (which is loosely based on Liang et al. 2009) that captures unresolved variability since we do not have sub-pixel information from RSP reflectance retrievals. We examined the dependence of RSP Re difference (between bi-spectral Re and polarimetric Re) on this derived cloud horizontal heterogeneity parameter, but we did not see clear dependence that was worth reporting. Instead, we focused our reporting of possible sub-pixel variability using the higher resolution HSRL-2 2Hz data as reported in Section 4.2.1.*

Regarding clear-sky contamination (Fig. 16) using the HRSL-2 lidar – what is the optical depth threshold used to define regions as cloud when estimating the cloud fraction? Is it comparable to that used for the bi-spectral RSP or MODIS retrievals? Perhaps if the HRSL-2 is very sensitive to cloud then its definition of cloud fraction within a cloudy segment is not meaningful for bi-spectral retrievals. What happens if you use a higher optical depth threshold?

*Response: The HSRL-2 aerosol/cloud classification does not use an optical depth threshold (Burton et al. 2010, Scarino et al. 2014), so we can't discuss it that way. Upon reviewing our original Fig. 2(d) and Fig. 2(e), we found a mismatch between the sampling of HSRL-2 and RSP cloud segments due to omitting the P-3's banking adjustments. This is now corrected for, and we have made the corresponding changes in the main text in Section 3.1 and Figure 2. From the updated Fig. 2(d) and 2(e), the statistics between the HSRL-2 and RSP derived cloud elements agree well. Also keep in mind that we are only calculating the cloud fraction from HSRL-2 for cloud segments that was detected by the RSP, in order to examine possible sub-pixel clear-sky contamination in the RSP Re retrievals. The higher resolution of the HSRL-2 2Hz over RSP adds to our confidence of the presence of sub-pixel clear-sky contamination in RSP cloudy segments. This is now discussed in more detail in the second paragraph of the "clear sky contamination" section of Section 4.2.1:*

*Line 669-677: "We used HSRL-2 2Hz data (~75 m horizontal resolution) to derive the cloud fraction (CF) for each RSP cloud element in an attempt to identify if RSP identified cloud elements could contain some unresolved clear sky. The higher resolution and greater sensitivity of the HSRL-2 relative to RSP plays out in Figure 2(d) and (e) cloud and clear length statistics, with the largest differences occurring for lengths < 200 m. The good agreement between the two instruments for in cloud lengths > 200 m and for CTH (Fig 2(c)) indicates similar sensitivity to clouds larger than the RSP footprint. Hence, using HSRL-2 CTH retrievals to derive a cloud fraction for RSP identified cloud elements should provide a relevant sub-pixel cloud fraction for the RSP retrievals. Still, this would be a minimum estimate of clear sky contamination in RSP cloud elements as the lidar itself may also contain unresolved clear sky."*

*Burton, S. P., Ferrare, R. A., Hostetler, C. A., Hair, J. W., Kittaka, C., Vaughan, M. A., Obland, M. D., Rogers, R. R., Cook, A. L., Harper, D. B., and Remer, L. A.: Using airborne high spectral resolution lidar data to evaluate combined active plus passive re-trievals of aerosol extinction profiles. J. Geophys. Res.-Atmos., 115, D00H15, doi:10.1029/2009JD012130, 2010.*

*Scarino, A. J., Obland, M. D., Fast, J. D., Burton, S. P., Ferrare, R. A., Hostetler, C. A., Berg, L. K., Lefer, B., Haman, C., Hair, J. W., Rogers, R. R., Butler, C., Cook, A. L., and Harper, D. B.: Comparison of mixed layer heights from airborne high spectral resolution lidar, ground-based measurements, and the WRF-Chem model during CalNex and CARES, Atmos. Chem. Phys., 14, 5547–5560, https://doi.org/10.5194/acp-14-5547-2014, 2014.*

The ASTER data (Fig. 12) does not seem to be used except to show that many of the clouds are smaller than 1km in size. Could it be used to estimate the effects of sub-pixel cloud heterogeneity (similar to what was done in Werner, 2018)? Since this could be a cause of the Re biases. Also, can you examine what the MODIS PCL retrievals estimate for Re?

*Response: We have modified the original text on the ASTER image and MODIS PCL to discuss quantitatively the MODIS PCL Re retrievals (Line 534-536 in Section 3.3.3 of revised manuscript). Using the same scene shown in Fig. 12 of the original manuscript, the MODIS PCL retrievals revealed that most of the Re were in the 10 to 30 µm range, with a median value of 18.5 µm, compared to the MODIS standard retrievals mostly in the 10 to 20 µm range, with a median value of 14 µm.*

*Unfortunately, we cannot follow Werner et al. (2018) because the ASTER SWIR channels were no longer functioning by the time CAMP²Ex took place. Still, one of Prof. Di Girolamo's graduate students is working on a similar approach that could still just use the ASTER VNIR channels, but that is work in progress that will not be part of this manuscript. We also note that while we did have good coordination with aircraft and MODIS during CAMP²Ex, this was not the case for ASTER in which there were only two good cases (the original Figure 12 being one of them). So we would not have a whole lot of MODIS-ASTER corrected retrievals to compare with RSP and in situ retrievals.*

I don't really see any evidence for this statement in the abstract (line 44): "3D radiative pathways appear to be the leading cause for the large positive biases in bi-spectral retrievals.". Where was this shown?

*Response: As noted in our response to the reviewer's first question, it came from a synthesis of analyses presented throughout the manuscript and summarized in our conclusion, part of which, for example, as:*

*Line 839 to Line 850: "Our analysis in Sect. 3.1 showed that most samples observed by the P-3 remote sensors came from small, optically thin, shallow clouds. The samples exhibit a large difference (~factor of 2) between RSP bi-spectral and polarimetric Re retrievals. For non-drizzling shallow clouds, in situ observations compare well against the RSP polarimetric retrievals, and show variability of within ~2 μm. For these non-drizzling shallow clouds, no in situ Re samples are as large as the RSP bi-spectral Re. Therefore, for the shallow clouds observed by RSP during CAMP²Ex, the long-held hypothesis of the presence of drizzle or vertical variations as major contributing factors to Re differences between bi-spectral and polarimetric retrievals could be rejected with near certainty. Also, as revealed by the HSRL-2 derived RSP cloud element cloud fraction, clear sky contamination only has very limited contribution (~1 μm) to the observed RSP Re differences. Thus, for the shallow, non-drizzling clouds, the evidence presented herein is strongly suggestive that the dominant cause for the differences between RSP polarimetric and bi-spectral Re observed during CAMP²Ex lies within 3-D radiative pathways that lead to large positive biases in bi-spectral retrievals of Re compared to polarimetric retrievals."*

Regarding the bias correction to the MODIS data – do the results here suggest that the lower bound of the correction is most suitable? Could this be added as a recommendation at the end?

*Response: The lower bound and the upper bounds in the bias-correction are measures that intend to bound the magnitude of Re bias, so technically they should work together in pairs. So our recommendation is to use both. Unfortunately, adding the lower bound estimate to all the figures made the figures very messy looking. So we decided to go with the upper bound to guide the discussion on the MODIS-bias correction results and add a summary of the lower bound results to the conclusion section. We now added to the conclusion an emphasis to use both and clarified a statement in the conclusion that led us to believe why the reviewer asked this question in the first place.*

Some of the figures seem a little bit redundant and the paper is very long in terms of the number of figures. Perhaps some of them could be removed, or put into an appendix or supplementary section? E.g., Figs. 13 and 14 don't seem to add much beyond what has already been shown. If you want to demonstrate that the CTH variability gets larger with height then there are more direct ways to do this. Fig. 15a seems the same as Fig. 3b and I'm not sure how much the other panels add for Fig. 15. The 4 figures for the different case studies start to get a bit repetitive too. Figures 6 and 7 are useful perhaps since Fig. 7 shows some interesting high Re values from the Lear Jet. But after that the story is similar with higher Re retrievals for the bi-spectral method. Maybe the rest of the case study figures (Figs. 8-11) could be put into an appendix?

*Response: The other reviewer also had the same comment. We have taken actions by reorganizing Section 3.3 and moved three of the four case studies (six figures) and their discussion to Supplemental Materials, removing Figure 13 and 17 in the original manuscript, and condensing Figure 15 in the original manuscript. The text was modified appropriately for these changes. The modifications to the text were just stylistic to ensure a logical flow and proper transitions. No qualitative or quantitative changes to discussion/results were taken.*

Specific comments
Fig. 2 – the caption should say whether the data is for all of flights for which data was captured and for only oceanic liquid clouds only.

*Response: We have added the details in the figure caption:*

*Line 370-374: "**Figure 2.** Probability distribution function (PDF, solid line) and cumulative distribution function (CDF, dash line) for cloud element mean values of (a) the RSP polarimetric and bi-spectral Re, (b) the RSP polarimetric and bi-spectral COT (c) CTH from RSP and HSRL-2, (d) cloud element transect length from RSP and HSRL-2, (e) clear segment length (between cloud elements) from RSP and HSRL-2, and (f) APR-3 W-band maximum reflectivity within a cloud element, using warm oceanic liquid clouds segments from all research flights."*

L357 – "The samples exhibited a large difference (a factor of ~2) between RSP bi-spectral and polarimetric Re retrievals, which is investigated further in the sections below."
- This sentence seems out of place at the end of a paragraph about radar reflectivity. I would argue that it is not necessary since it is clear by now that the paper is focuses on Re. But if you want to keep this to lead into the focus on Re then it would be better in its own paragraph and would need some rewording.

*Response: We agree and have removed this sentence.*

"Cloud bow COT" – this is used a few times and is only explained in the Results section rather than in the Methods – a short description of how this is done would be better placed in Section 2.1.1 then in the Results section. I also think that just calling it the RSP COT would be clearer (after you have defined what this means). It would then be more aligned with how you describe the Re from the RSP.

*Response: Agreed. We have changed it to RSP COT, and added this in Section 2.1.1:*

*Line 165-167: "For COT, the standard COT product from RSP is retrieved using total reflectances and polarimetric Re; RSP also reports the bi-spectral retrieved COT."*

Fig. 4h – it's a bit hard to see the CTH dots – would this be better as a separate timeseries and maybe with a line as well as dots? This could perhaps be overlaid on the radar reflectivity plot.

*Response: Agreed. We moved the CTHs to the radar reflectivity plot and it is much clearer to identify.*

Fig.7 – it's interesting that here the RSP-pol and adjusted MODIS values agree better with the in-situ values at lower altitudes, but at higher altitudes the bispectral and non-adjusted MODIS values do. You should mention this even though you suggest that the Lear jet samples were biased high due to flying through a raining turret (i.e., sampling bias – this could be spelled out a bit more clearly in the text). Perhaps other explanations could be a potential role of cloud top entrainment causing the RSP-pol Re retrievals to be low (since they sample very close to cloud top), whereas deeper in the clouds (in-situ and bi-spec retrievals) the Re values are higher?

*Response: Agreed. However, as past studies through simulations and observations have shown, the impact of entrainment and mixing to the horizontal variation of Re from the exterior to the interior of a cloud is small (e.g., Khain et al. 2019; Gerber et al. 2008). Thus it is unlikely that the large difference we see between the bi-spectral Re and the Re retrieved from other techniques can be explained by entrainment and mixing alone.*

*As mentioned above we have reorganized Section 3.3 and moved three of the four case studies (which includes Fig.7) and their discussion to Supplemental Materials. So we have modified the discussion around Fig. 5 to reflect this point and clarify our statement.*

*Line 456-460: "While acknowledging the different sensitivity to cloud exterior and in-cloud microphysics for these different techniques, observations and simulations have shown for shallow cumulus clouds that low variation of Re (~10%) exist between the exterior and interior of clouds at a given altitude (e.g., Khain et al. 2019; Gerber et al. 2008). Thus, we primarily focus on accounting for the systematic variation with altitude."*

*And also:*

*Line 478-479: "Apart from sampling differences, some of the variability between in situ Re and the remotely sensed Re may also be due to entrainment and mixing (e.g., Gerber et al. 2008)."*

*Khain, P., Heiblum, R., Blahak, U., Levi, Y., Muskatel, H., Vadislavsky, E., Altaratz, O., Koren, I., Dagan, G., Shpund, J., & Khain, A.: Parameterization of Vertical Profiles of Governing Microphysical Parameters of Shallow Cumulus Cloud Ensembles Using LES with Bin Microphysics, J Atmos. Sci., 76(2), 533-560. https://doi.org/10.1175/JAS-D-18-0046.1, 2019.*

*Gerber, H., Frick, G., Jensen, J., and Hudson, J.: Entrainment, mixing, and microphysics in trade-wind cumulus, J. Meteorol. Soc. Jpn., 86, 87–106. https://doi.org/10.2151/jmsj.86A.87, 2008.*

Figs. 4, 8 and 10 – the captions should say which flight the plots were for (RFxxx).

*Response: We have added them to the captions.*

Case study introductory paragraphs – i.e., Sections 3.3.1 – 3.3.4. These should mention the relevant overview figures early on (i.e., Figs. 4,6 8 and 10) when giving the general description of the cases.

*Response: Our case study introductory paragraph Line (423-438) starts with discussing the relevant features in the overview figures, and we have also moved three of the four case studies into Supplemental Materials.*

Fig. 12 – what happens if you use the PCL retrievals from MODIS?

*Response: We have addressed this question above in the response to the reviewer's third question.*

Figs. 13 and 14 – are these necessary? I'm not sure that they add much beyond what has already been shown. If you want to demonstrate that the CTH variability gets larger with height then there are more direct ways to do this.

*Response: As noted above, we have decided to remove Fig. 13 (of the original manuscript) and keep Fig. 14 (of the original manuscript). Figure 14 not only shows CTH variability with height, but it also shows that 1) the Re variability at a cloud element level is smaller for the polarimetric Re, and 2) mean Re values are smaller for the polarimetric Re. Figure 14 shows these two points are consistent across all RF segments that sampled warm cumulus clouds. Since it's a nice graphical summary of all valid cloud elements sampled over all the flights, it may help inform interested readers wishing to explore further which flights to focus on.*

Fig. 15a – how is this different to Fig. 3b? Do panels b and c add much more? And would they be better as density maps instead of scatter plots as in Fig. 3?

*Response: Figure 15(a) is using cloud elements, while Fig. 3(b) are original full resolution samples. The amount of data is very limited, and density maps are very coarse that do not reveal the main features. But we did take the opportunity to combine Fig. 15(b) and (c) into a single panel by introducing color. This made the figure easier to interpret and reduced the number of panels in Fig. 15 from three to two.*

L743 – "to derive the cloud fraction (CF) for each RSP cloud element as follows: For RSP cloud elements that have HSRL-2 CTH retrievals, the HSRL-2 cloud fraction for a RSP cloud element is defined as the number of valid HSRL-2 CTH retrievals divided by the total number of HSRL-2 CTH retrievals within the RSP cloud element"
- This is a little confusing – do you mean that you divide by the total number of *attempted* HSRL-2 CTH retrievals?

*Response: Yes, we divide the total number of valid HSRL-2 CTH retrievals with total number of HSRL-2 data samples for each cloud element. The sentence has been*

*reworked for clarity, along with the discussion of HSRL-2 sensitivity in response to the reviewer's second comment above.*

L756 – "there are no samples for CF < 0.5 (hence no red dashed histogram)" – it would be good to mention this in the caption of the figure (16) too.

**Response:** *We have now added this to the caption.*

Typos / grammar

**Response:** *For all of the typo/grammar suggestions from L41 to L846, we want to thank the reviewer for identifying each of them. We have fixed all the typo/grammar problems as suggested.*

L41 – "RSP bi-spectral Re shows larger relative values compared to RSP polarimetric Re for smaller and optically thinner clouds." – this doesn't quite get across the result that the bias is worse for smaller optical depths. I recommend:- "The overestimate of Re from the RSP bi-spectral method relative to Re from the polarimetric RSP method increased as cloud size and optical depth reduced."

**Response:** *Agreed. We have changed it to the following: "The overestimates of Re from RSP bi-spectral technique compared to polarimetric technique increased as cloud size and cloud optical depth decreased."*

L69 – "(hence, 1-D radiative transfer as the forward model used in this retrieval)," – better as "(hence, 1-D radiative transfer is used as the forward model in this retrieval)," I think.

**Response:** *We have changed it accordingly.*

L142 – "because the overpass time of the latter two sensors occurs in the afternoon when cirrus is more frequent and when the aircraft was returning to base that did not have favourable samplings.". Better as :- "because the overpass time of the latter two sensors occurs in the afternoon when cirrus is more frequent and when the aircraft was returning to base; therefore, the sampling was not favourable."

**Response:** *We have changed it accordingly.*

L174 – "extend its evaluation in cumulus cloud fields" -> "extend its evaluation to cumulus cloud fields"

**Response:** *We have changed it accordingly.*

L179 – "Mean and standard deviations of retrieved quantities belonging to a cloud element is computed." -> "Means and standard deviations of retrieved quantities are computed for cloud elements."

*Response: We have changed it accordingly.*

L200 – "Only drop size distribution with" -> "Only drop size distributions with" + "temperature" -> "temperatures"

*Response: We have changed it accordingly.*

L211 – "The median differences within 1 μm" – "The median differences were within 1 μm".

*Response: We have changed it accordingly.*

L256 – "pixel" -> "pixels".

*Response: We have changed it accordingly.*

L268 – "to retrieve COT at MISR 9 view angles" -> "to retrieve COT at MISR's 9 view angles"

*Response: We have changed it accordingly.*

L267 – "As a continuation of Liang et al. (2015), fused MISR L1B radiance data and MODIS L2 cloud Re were used to retrieve COT at MISR 9 view angles.".
Better as :- "As a continuation of Liang et al. (2015), Fu et al. (2019) fused MISR L1B radiance data and MODIS L2 cloud Re to retrieve COT at MISR's 9 view angles." (Since otherwise it sounds like this was done in the paper under review here).

*Response: We have changed it accordingly.*

L330 – "The median COT is 3.5 for the cloud bow (COT retrieved using total reflectances and polarimetric Re) and 4.2 for the bi-spectral." – add "retrievals" at the end.

*Response: We have changed it accordingly.*

L440 – "over which little overall change in the cloud field" – add "occurs" after this.

*Response: We have changed it accordingly.*

L441 – "Here, the Re retrievals were sorted into 250 m CTH bins." – this makes it sound like you are referring to a figure or a result that has already been introduced. You could just say "We sorted the Re retrievals into 250 m CTH bins."

*Response: We have changed it accordingly.*

L581 – "Apart from these clouds being sub-pixel to MODIS retrievals cannot resolve" -> "Apart from these clouds being sub-pixel to MODIS retrievals so that MODIS cannot resolve them"

*Response: We have removed this sentence in the manuscript.*

L609 – "As mentioned in Sect. 2.2, we acknowledge the difference between the MODIS 2.1 μm channel to the RSP 2.26 610 μm channel, we acknowledge that the differences in the two wavelengths and we do not expect the Re retrieved from MODIS and RSP bi-spectral to have the exact same bias" -> – "As mentioned in Sect. 2.2, we acknowledge the difference between the MODIS 2.1 μm channel to the RSP 2.26 610 μm channel  so we do not expect the Re retrieved from MODIS and RSP bi-spectral to have the exact same bias"

*Response: We have changed it accordingly.*

L629 – "that can be" -> "than can be".

*Response: We have changed it accordingly.*

L660 – "A prominent feature in Fig. 13 is the much-improved correlation between RSP polarimetric Re and CTH means, with a linear correlation coefficient of 0.72, compared to the correlation coefficient between RSP bi-spectral Re and CTH means of 0.24." -> "A prominent feature in Fig. 13 is the much-improved correlation between the RSP polarimetric Re and CTH means (linear correlation coefficient, r, of 0.72) compared to between the RSP bi-spectral Re and CTH means (r=0.24)."

*Response: We have removed Fig. 13 along with the relevant discussions of Fig. 13, and updated this discussion with the new Fig. 7 in the revised manuscript:*

*Line 594-596: "A prominent feature in Fig. 7 is the high degree of correlation between RSP polarimetric Re and CTH means (linear correlation coefficient, r, of 0.64 averaged across all 12 cases), compared to the correlation between RSP bi-spectral Re and CTH means (r = 0.18 averaged across all 12 cases."*

L844 – "Also as noted in Sect. 3, are associated with aircraft penetration of deeper clouds not near the same CTH level that contain drizzle." -> "Also as noted in Sect. 3, they are associated with the penetration of deeper clouds at altitudes different to the CTH level observed by the remote sensing, with the clouds tending to contain drizzle."

*Response: We have changed it accordingly.*

L846 – "Overall, RSP polarimetric Re and bias-adjusted MODIS Re, the Learjet, P-3 in situ indicated" -> "Overall, Re observations from the RSP polarimetric, the bias-adjusted MODIS, the Learjet in situ and the P-3 in situ techniques indicated…"

*Response: We have changed it accordingly.*

L864 – "have closer median Re values within their differences within 1 to 2 μm," – not clear what you mean by "within their differences"?

*Response: We have changed it to:*
*Line 784: "… in situ have closer median Re values **that are** within 1 to 2 μm".*

References
Liang, L., Di Girolamo, L., & Platnick, S. (2009). View-angle consistency in reflectance, optical thickness and spherical albedo of marine water-clouds over the northeastern Pacific through MISR-MODIS fusion. Geophysical Research Letters, 36, L09811. https://doi.org/10.1029/2008GL037124
Zhang, Z., Werner, F., Cho, H. M., Wind, G., Platnick, S., Ackerman, A. S., et al. (2016). A framework based on 2-D Taylor expansion for quantifying the impacts of subpixel reflectance variance and covariance on cloud optical thickness and effective radius retrievals based on the bispectral method. Journal of Geophysical Research: Atmospheres, 121, 7007–7025. https://doi.org/10.1002/2016JD024837
Werner, F., Zhang, Z., Wind, G., Miller, D., & Platnick, S. (2018). Quantifying the impacts of subpixel reflectance variability on cloud optical thickness and effective radius retrievals based on high-resolution ASTER observations. Journal of Geophysical Research: Atmospheres. 123, 1–20. https://doi.org/10.1002/2017JD027916

**Reviewer 2:**

The manuscript documents the validation of MODIS cloud droplet effective radius (Re) during CAMP2eX using in-situ and remotely sensed data. This work is highly relevant as assessments of satellite retrievals for shallow cumulus clouds, where clouds tend to be small and precipitation is frequent, are scarce. While the expectation is that satellite Re significantly departs from observations, having data to support this assertion is essential. This is why Fu et al. is a key study to understand the limitations of MODIS-like retrievals and the implications for climate studies. The authors had to circumvent multiple issues associated with the fact that the sampling strategy was not designed for validating satellite retrievals of cloud properties. I commend the efforts of the authors and the attempts to understand the satellite positive biases. Given the lack of data collocation, the authors mainly report biases, whereas spatial correlations and co-variability are difficult to compute with the CAMP2EX dataset.

While the main message is straightforward and the discussion easy to understand, the number of figures is excessive, making the manuscript a bit difficult to follow. I am providing below a number of suggestions to help polish a few sections and make the manuscript more concise.

*Response: We are glad the reviewer recognized the need for this study, the efforts that we went through to put it together, and the straightforward message we are conveying. We address all of the reviewer's comments below.*

4, 6, 8, 10: The inclusion of multiple panel figures is repetitive and the information conveyed is quite similar. What matters is: a) clouds have small sizes, precipitation was frequently observed (contrary to what is stated in the manuscript), and MODIS Re is overestimated. I find it particularly hard to understand the excessive interest in cloud top height. From a remote sensing perspective, as long as the focus in on boundary layer clouds, CTH does not matter. The only reason to justify

the use of CTH is for exploring the cloud topography and the associated 3D radiative effects. Because no directs comparisons are made between RSP/insitu data and MODIS, my suggestion would be to include PDF for Re, optical depth, radar reflectivity (more below). To convey information about cloud coverage, one could easily compute cloud fraction from the HSRL using, for instance, 30s segments and this new parameter can be depicted in a PDF. MODIS Re maps are informative so the authors should consider keeping them. In sum, 4 multi-panel figures can be combined into only one: an upper panel for the MODIS maps, and a lower panel with multiple PDFs for each RF.

*Response: Reviewer 1 also thought the number of figures was excessive, and we have taken action to move three of the four case studies (which included Fig. 6, 8, and 10 in the original manuscript) to Supplemental Materials. See our response to Reviewer 1 for more details.*

*The reviewer is correct that frequent drizzle was observed, and we have modified the manuscript throughout to reflect this fact, including our discussion around Figure 2 and Section 4.2.2.*

*True, from a remote sensing of Re_2.1 perspective, the CTH is not that important in 1-D when the cloud is truly plane-parallel. However, for real 3-D clouds, CTH does play a role in the Re bias (e.g., consider all those papers such as Coakley and Davies 1986 discussing the role of cloud aspect ratio on 3-D effect on the emerging radiation field). Thus CTH and cloud horizontal extent must be shown, which are conveyed in panel (g) of these figures. Keeping the other panels also convey spatial context and spatial variability that are also important to 3-D effects, which would get lost when summarized into PDFs. Thus, we are keeping the panels for these figures, but as noted above, moving several of the figures to the Supplemental Materials.*

Bi-spectral RSP Re is not the same as MODIS. I understand why the authors are using the Bi-spectral RSP retrievals, but they need to keep in mind: 1) at the RSP footprint size, 3D radiative transfer effects are going to be substantial, 2) On the other hand, clear-sky (cloud free) contamination is going to be a much bigger problem for MODIS than RSP, so it is not correct to assume that absolute uncertainties/biases derived from RSP are representative of MODIS, 3) viewing geometry are quite different (see comment #4)

*Response: We agree with the reviewer, but nowhere in the manuscript did we assume that the absolute uncertainties/biases derived from RSP are representative of MODIS. It just so happens that their overall behavior relative to the non-bi-spectral methods are roughly the same. And we do acknowledge these factors throughout the manuscript when making comparisons, for example as summarized in Section 3.3.2 of the revised manuscript:*

*Line 503-507: "In our analysis, other contributing factors that may impact the differences between RSP 2.26 μm bi-spectral Re and MODIS 2.1 μm Re include (1) sampling differences (2) channel differences in the face of vertical and horizontal variations in cloud optical properties, and (3)*

*pixel size difference in the face of 3-D variations. Despite these factors, the RSP 2.26 μm bi-spectral Re and MODIS 2.1 μm Re have very similar behavior, exhibiting a large positive bias and much greater variability in Re relative to the other techniques."*

3.7-um Re: It has been shown in several studies that Re derived from the 3.7-um channel is less sensitive to spatial inhomogeneities and 3D radiative effects. Moreover the vertical photon penetration is confined to the uppermost cloud layer with optical depth of 2 or less. 3.7-um Re is also adopted by CERES for deriving radiative fluxes product. Given the increasing use of the 3.7um Re, I would like to encourage the authors to take a look at this product.

**Response:** *Correct, the differences between MODIS Re_2.1 and MODIS Re_3.7 has been extensively studied in other papers (e.g., Zhang and Platnick 2011, Fu et al. 2019). Still, as part of this CAMP2Ex study, we did examine the 3.7 channel MODIS Re as well, but didn't conclude anything different than what was found in these others studies. This is now discussed at the end of Section 2.2:*

*Line 266-269: "We also examined the MODIS Re product derived using its 3.7 μm channel and found the differences between the Re products derived from MODIS 2.1 μm and 3.7 μm channels to be consistent with what was reported in previous studies (e.g., Zhang and Platnick 2011; Fu et al. 2019). So they are not included in the figures and tables in the following sections. However, we provide a brief summary of these differences at the end of Section 4.3."*

*And at the end of Section 4.3, we added:*

*Line 792-798: "While we have been using the MODIS Re product derived from its 2.1 μm channel, we also examined the MODIS Re derived from its 3.7 μm channel. For MODIS Re derived from its 3.7 μm channel, the median, mode, and mean ± standard deviation are 14.3, 12.5, and 15 ± 4.8 μm for warm clouds over all research flights. Comparing these values to those in Table 5, we see that the MODIS 3.7 μm retrievals are ~ 2 to 3 μm smaller than MODIS 2.1 μm retrievals, which is thought to be due a combination of less impact from 3-D effects and differences in the vertical weighting of the two channels (e.g., Zhang and Platnick 2011). The MODIS 3.7 μm retrievals are still larger than the RSP polarimetric retrievals by ~3 to 5 μm."*

My understanding from Liang et al. and Fu et al. is that the Re correction is for removing the odd behavior of the retrievals near the rainbow. In other words, the correction only applies to satellite scattering angles around 138°-140°. If so, information about satellite scattering angle should be provided. Is it possibly that the bias in MODIS is primarily explained by the rainbow effect? Similarly, if angles around 140° are observed by Terra MODIS, I am wondering whether Aqua angles differ from Terra. If so, then Aqua Re < Terra Re…possibly, the diurnal cycle of clouds plays a roles…but it would be interesting to investigate Re differences in terms of scattering angle. In any case, the readers need more information about the correction method and why the method is applicable to CAMP2EX.

**Response:** *The reviewer's understanding of the bias-correction isn't quite correct. The method of Liang et al. exploits information in the cloud bow scattering angles observed by MISR to characterize the MODIS Re bias regardless of the MODIS scattering angles.*

*The method of Fu et al. takes this one step further by developing regression relationships between the MISR-MODIS estimated MODIS Re bias and retrieved optical depth, heterogeneity, region, and time of year. We have modified the description of the correction method as followed in Section 2.3 to clarify this point:*

*Line 271-289: "The MODIS Re bias estimates presented in Fu et al. (2019) are also evaluated by comparing against the CAMP²Ex dataset. As a continuation of Liang et al. (2015), Fu et al. (2019) fused MISR L1B radiance data and MODIS L2 cloud Re to retrieve COT at MISR's 9 view angles. Liang et al. (2015) revealed that the COT retrievals show a local minimum around the cloud-bow scattering direction (~140°), and this feature was prominent throughout both MODIS cloud COT values and COT retrieved from MISR. They showed that this minimum was attributed to an overestimate in the MODIS Re product, and that the value of Re bias could be estimated. The local minimum of COT was prevalent in a multiple-year climatology carefully stratified by scattering angle, latitude and solar zenith angle rather than apparent at a ~1-km pixel resolution within a Level 2 MODIS granule because of large spatial variability in scattering angle, solar zenith angle and cloud heterogeneity within a Level 2 granule. To correct for the Re bias, Fu et al. (2019) used 8 years of MISR and MODIS data, and further stratified by MISR nadir τ and cloud heterogeneity to produce climatology estimates of corrected MODIS Re between 60° N and 60° S globally at 2.5° resolution for the months of January and July. In the current study, we apply the July regional correction factors from Fu et al. (2019) at 2.5° to the MODIS L2 granules over the CAMP²Ex domain to better compare with Re derived from other techniques under similar seasonal conditions. This allows one to test the robustness of the correction. The average of the July correction factors over the CAMP²Ex domain is ~ 0.6. The correction factors over this region range from 0.25 to 0.97 depending on latitude, τ and cloud heterogeneity. We are interested in evaluating the capability of regional bias corrections to capture the actual variability at its original resolution (i.e., MODIS 1 km retrieval) as we compare to field measurements from CAMP²Ex.*

Cloud top height (CTH): I cannot understand the rationale for performing the analysis as a function of CTH. I find Fig. 5,7,9, and 11 somewhat misleading because the remote sensors don't retrieve vertical profiles as retrieved cloud properties are representative of those near the cloud top. Moreover, equating remote sensing samples (derived from clouds with different top heights) with the cloud vertical profile is problematic, especially for precipitating shallow cumulus clouds. Another pitfall of the method is that MODIS CTH is likely biased.

**Response:** *We thought it would be understood from the remote sensing context that we are dealing not with the vertical variation of Re in the cloud, but rather the variation of Re with cloud top height. We have now clarified this point in Section 3.3.1:*

*Line 452-454: "The Re mean and standard deviation are computed for each height bin as a means of comparing remote sensing techniques' ability to capture the variations of Re with cloud top height, which is important when using the data for understanding cloud processes."*

*Past observation and simulation studies has shown for shallow cumulus clouds that there is low variation of Re at a given altitude (e.g., Khain et al. 2019; Gerber et al. 2008). Rather than focusing on this small variation, we are more interested in accounting for the Re variation with altitude, since Re has systematic dependence on temperature.*

*We have also added the following discussion in the manuscript:*

*Line 456-462: "While acknowledging the different sensitivity to cloud exterior and in-cloud microphysics for these different techniques, observations and simulations have shown, for shallow cumulus clouds that low variation of Re (~10%) exist between the exterior and interior of clouds at a given altitude (e.g., Khain et al. 2019; Gerber et al. 2008). Thus, we primarily focus on accounting for the systematic variation with altitude. To do this, we binned Re retrievals from all 5 techniques (P-3 in situ, RSP polarimetric, RSP bi-spectral, MODIS bi-spectral, and bias-adjusted MODIS) separately as a function of binned CTH/altitudes."*

Precipitation: A threshold of 0dBz for precipitation detection is too high (Comstock et al., 2004 (https://doi.org/10.1256/qj.03.187). If the goal is to determine the effect of the bimodal distribution on the satellite retrievals, then, instead of using the maximum reflectivity of the column, one should limit the analysis to the upper cloud layer (e.g. a 100-m layer from the cloud top).

**Response:** *Agreed. As noted in the response to Reviewer 2's second comment above, we have modified the manuscript throughout to reflect this fact, including our discussion around Figure 2 and Section 4.2.2. We also took the reviewer's "upper cloud layer" suggestion into account and modified the text and figure in Section 4.2.2 accordingly.*

Cloud fraction: I speculate that the effect of cloud fraction (clear-sky contamination) on MODIS retrievals is substantial. One could test this hypothesis by estimating cloud fraction using ASTER or the 250m MODIS visible channel.

**Response:** *Agreed. Using the scene in Fig. 12, we did compare the cloud fraction from the 15 m resolution ASTER imagery for MODIS 1km retrievals. For the domain displayed in Fig 12(b), based on a simple threshold ASTER cloud mask from the ASTER imagery, ASTER cloud fractions were derived from the ratio between cloudy pixels vs. total number of pixels. We found that for all the MODIS 1km pixels that reported a retrieved Re, the ASTER cloud fraction ranged between 0.25 to 0.62. It indeed reflects that sub-pixel heterogeneity is quite substantial for the MODIS retrievals. We have modified the text in the revised Section 3.3.3 to reflect these results.*

Other comments:

Line 460-461: From Fig 4, I could not see any correlation between "drizzle" and larger Re.

**Response:** *As noted earlier, we moved the overlay of the CTH in the RSP Re to the APR-3 reflectivity, making the RSP Re plot less busy. The correlation is now more evident (and a scatter plot, not shown, between the two confirms this), with larger Re for both polarimetric and bi-spectral Re increasing with increasing APR-3 W-band reflectivity.*

Lines 628-629. While I need to read Miller et al., the same factors could also yield underestimations of Re.

*Response: We clarified in the text that this is for low optical depth.*

Line 657: I don't find Fig. 2c a convincing validation of RSP CTH, especially for boundary layer clouds.

*Response: Figure 2(c) is a general description of the RSP and HSRL-2 sampled cloud top heights; it is not stated that this is a validation of RSP CTH, although the statistics in cloud top heights between the two instruments are in excellent agreement.*

Line 707: Is the rainbow angle in the backscatter direction? If so, then RSP retrievals are always sampled at the backscatter direction. This means that the expected behavior is an underestimation of particle size (illumination effect).

*Response: This sentence is specific to the RSP bi-spectral retrieval (nadir) and not the RSP polarimetric retrieval (rainbow). As stated in Line 632, the nadir view was never taken near the backscatter direction.*

Line 771-772. Overestimation are only observed for high SZA, otherwise, optical depth is underestimated. Please revise the sentence.

*Response: The sentence did not say that the bias in retrieved COT is either overestimated or underestimated, simply that the biases are larger.*

Figure 15: I see a relationship between delta Re and COT, but no relationship can be observed between transect length and delta Re or COT (Fig. 15 b and c). Does the yaxis of Fig. 15 represent optical depth?

*Response: We combined the original Fig. 15(b) and (c) into a single panel by introducing color (Now Fig. 8(b) in the updated manuscript). This made the figure easier to interpret and reduced the number of panels from three to two.*

---

## Referee Report (RR1)

**Review of Fu, et al., 2022. (v3)**

Section 4.2 discusses the effects of 3D radiative effects, clear-sky contamination and drizzle, but not for sub-pixel heterogeneity (as in Zhang, 2016 – now referenced in the new manuscript). In the authour's response you say that some attempts to do this were tried, but appropriate sub-pixel variability data was not available. Perhaps this could be explained as an extra section in Section 4.2.1 with some discussion that sub-pixel variability therefore remains as a candidate for the cause of the Re overestimate from the bi-spectral approach?

Fig. 8a – this would be better as a density plot rather than a scatter plot (i.e., with colours showing the frequency within each x-y bin) since the points become cluttered at low COT.

Original review comment :-

I don't really see any evidence for this statement in the abstract (line 44): "3D radiative pathways appear to be the leading cause for the large positive biases in bi-spectral retrievals.". Where was this shown?

Response: As noted in our response to the reviewer's first question, it came from a synthesis of analyses presented throughout the manuscript and summarized in our conclusion, part of which, for example, as:

Line 839 to Line 850: "Our analysis in Sect. 3.1 showed that most samples observed by the P-3 remote sensors came from small, optically thin, shallow clouds. The samples exhibit a large difference (~factor of 2) between RSP bi-spectral and polarimetric Re retrievals. For non-drizzling shallow clouds, in situ observations compare well against the RSP polarimetric retrievals, and show variability of within ~2  $\mu$ m. For these non-drizzling shallow clouds, no in situ Re samples are as large as the RSP bi-spectral Re. Therefore, for the shallow clouds observed by RSP during CAMP2Ex, the long-held hypothesis of the presence of drizzle or vertical variations as major contributing factors to Re differences between bi-spectral and polarimetric retrievals could be rejected with near certainty. Also, as revealed by the HSRL-2 derived RSP cloud element cloud fraction, clear sky contamination only has very limited contribution (~1  $\mu$ m) to the observed RSP Re differences. Thus, for the shallow, non-drizzling clouds, the evidence presented herein is strongly suggestive that the dominant cause for the differences between RSP polarimetric and bispectral

Re observed during CAMP2Ex lies within 3-D radiative pathways that lead to large positive biases in bi-spectral retrievals of Re compared to polarimetric retrievals."

I'm afraid that I don't agree with this conclusion. The main evidence presented seems to be that the highest Re biases are at low COD. I agree that this does look similar to the effects of 3D radiative biases as presented in Fig. 5 of Marshak (2006). However, it is not hard evidence and the possibility of other explanations (e.g., sub-pixel variability - see above) should be acknowledged unless more substantial evidence can be presented. Also, as noted by the other reviewer, at the high resolution of the aircraft 3D radiative effects are more likely to be resolved than sub-pixel and hence 3D radiative effects may be larger than for MODIS - it would be good to discuss this in this section too.

There are still a few references to "cloud bow" COT, etc. in Section 3.2 - as discussed earlier it would be good to remove these to avoid confusion.

The newly added text contains a few grammatical errors that hopefully the type editors will pick up - or it might be useful to get it checked over if not.

---

## Author Response (AR2)

We would like to thank the editor again for handling the review process for our manuscript. We also thank Reviewer 1 for the careful review of the revised manuscript. We believe that the remaining minor concern of Reviewer 1 is simply due to semantics (basically, what is meant by "3-D effects"). We have re-worded the parts in question and addressed all comments. Our responses are listed below in blue italics. Comments from the reviewers are in upright black plain texts. The line numbers in our responses are referring to the line numbers in the revised manuscript.

**Review of Fu, et al., 2022. (v3)**

Section 4.2 discusses the effects of 3D radiative effects, clear-sky contamination and drizzle, but not for sub-pixel heterogeneity (as in Zhang, 2016 – now referenced in the new manuscript). In the authour's response you say that some attempts to do this were tried, but appropriate sub-pixel variability data was not available. Perhaps this could be explained as an extra section in Section 4.2.1 with some discussion that sub-pixel variability therefore remains as a candidate for the cause of the Re overestimate from the bi-spectral approach?

**Response:**

We agree. Following the suggestion, we have now added discussion in Section 4.2.1 and the conclusion section as followed:

Line 656 to Line 675: "It has been pointed out by several studies that the bi-spectral Re retrieval has a sensitivity to instrument resolution due to a) the nonlinear relationship between VNIR and SWIR reflectances and the COT and Re and b) the presence of variability in cloudy reflectances at all scales (e.g., Marshak et al. 2006; Zhang et al. 2012; Zhang et al. 2016; Werner et al. 2018a). An important example of this effect is clear sky contamination, in which cloudy radiances and clear sky radiances are both present within the field of view (FOV) of the sensor. The presence of this sub-pixel clear sky can cause Re overestimates of up to 41% when decreasing instrument resolution from 30 m to 1 km (Werner et al. 2018b). The bias in the bi-spectral retrieval due to clear sky contamination decreases monotonically as instrument resolution increases. This is due to the applicability of an independent column approximation as a model of the variability within the FOV due to the negligible atmospheric and surface scattering contributions over ocean surfaces to VNIR and SWIR radiance. Other reflectance variations within cloudy portions of an instrument FOV also cause a sensitivity of the bispectral retrieval to instrument resolution, though this is typically smaller, being 1 - 3 µm (Zhang et al. 2016; Werner et al. 2018a). In this case, increasing instrument resolution does not necessarily cause a monotonic reduction in retrieval bias (e.g., Davis et al. 1997; Zhang et al. 2012). That is because the relationship between the heterogeneity of the optical and microphysical properties (e.g., *Re)* within the cloud and the radiance field is governed by 3-D radiative transfer, not an independent column approximation. Note that polarimetric retrievals are only weakly sensitive to instrument resolution as they are largely unaffected by clear sky contamination (e.g., Miller et al. 2018; Shang et al. 2015). Based on these considerations, we assess the sensitivity of our bias estimate in the bispectral Re due to the relative coarse resolution of RSP by investigating the impact of clear sky contamination using the higher resolution HSRL-2 lidar. We may then attribute the remaining bias in Re to the expression of cloud heterogeneity and 3-D radiative transfer, whether this occurs at resolved or unresolved (i.e., sub-pixel to HSRL-2) scales."

**And:**

Line 856 to Line 859: "Thus, for the shallow, non-drizzling clouds, the evidence presented herein is strongly suggestive that the dominant cause for the differences between RSP polarimetric and bi-spectral Re observed during CAMP2Ex is due to 3-D radiative transfer and cloud heterogeneity (both resolved and unresolved by RSP) effects that lead to large positive biases in bi-spectral retrievals of

Re compared to polarimetric retrievals."

Fig. 8a – this would be better as a density plot rather than a scatter plot (i.e., with colours showing the frequency within each x-y bin) since the points become cluttered at low COT.

**Response:**

We experimented with both density plot and scatter plot when we first made the figures. The density plot is attached here, we have replaced Fig. 8(a) with this density plot in the main text.

*Figure* 8(*a*). *Density plot of cloud element mean Re difference ((bi-spectral – polarimetric Re) vs. mean COT.*

Original review comment :- (the original reviewer comment in R1 is in red, the original response in light blue)

I don't really see any evidence for this statement in the abstract (line 44): "3D radiative pathways appear to be the leading cause for the large positive biases in bi-spectral retrievals.". Where was this shown?

Response: As noted in our response to the reviewer's first question, it came from a synthesis of analyses presented throughout the manuscript and summarized in our conclusion, part of which, for example, as:

Line 839 to Line 850: "Our analysis in Sect. 3.1 showed that most samples observed by the P-3 remote sensors came from small, optically thin, shallow clouds. The samples exhibit a large difference (~factor of 2) between RSP bi-spectral and polarimetric Re retrievals. For non-drizzling shallow clouds, in situ observations compare well against the RSP polarimetric retrievals, and show variability of within ~2  $\mu$ m. For these non-drizzling shallow clouds, no in situ Re samples are as large as the RSP bi-spectral Re. Therefore, for the shallow clouds observed by RSP during CAMP2Ex, the long-held hypothesis of the presence of drizzle or vertical variations as major contributing factors to Re differences between bi-spectral and polarimetric retrievals could be rejected with near certainty. Also, as revealed by the HSRL-2 derived RSP cloud element cloud fraction, clear sky contamination only has very limited contribution (~1  $\mu$ m) to the observed RSP Re differences. Thus, for the shallow, non-drizzling clouds, the evidence presented herein is strongly suggestive that the dominant cause for the differences between RSP polarimetric and bispectral Re observed during CAMP2Ex lies within 3-D radiative pathways that lead to largepositive biases in bi-spectral retrievals of Re compared to polarimetric retrievals."

I'm afraid that I don't agree with this conclusion. The main evidence presented seems to be that the

highest Re biases are at low COD. I agree that this does look similar to the effects of 3D radiative biases as presented in Fig. 5 of Marshak (2006). However, it is not hard evidence and the possibility of other explanations (e.g., sub-pixel variability - see above) should be acknowledged unless more substantial evidence can be presented. Also, as noted by the other reviewer, at the high resolution of the aircraft 3D radiative effects are more likely to be resolved than sub-pixel and hence 3D radiative effects may be larger than for MODIS - it would be good to discuss this in this section too.

**Response:**

We agree. We believe that the concern here deals with semantics – basically our use of "3-D radiative pathways" includes pathways involving sub-pixel cloud heterogeneity. For clarity, we changed "3-D radiative pathways" to "3-D radiative transfer and cloud heterogeneity" in both the abstract and the conclusion. By calling it "3-D radiative transfer and cloud heterogeneity", we include all effects of 3-D radiative transfer that are both "external" (such as cloud shape) and "internal" (i.e., microphysical variability) cloud heterogeneity, whether they are resolvable or not by the measurement. The paragraph added to Section 4.2 in response to the reviewer's first question also clarifies this.

There are still a few references to "cloud bow" COT, etc. in Section 3.2 - as discussed earlier it wouldbe good to remove these to avoid confusion.

**Response:**

We have removed all "cloud bow" and they are now "COT" as suggested.

The newly added text contains a few grammatical errors that hopefully the type editors will pick up - or it might be useful to get it checked over if not.

**Response:**

We have gone through the entire text and found several grammatical errors/typos and we have now corrected them.